# PFDiff: Training-free Acceleration of Diffusion Models through the Gradient Guidance of Past and Future

## Abstract

Diffusion Probabilistic Models (DPMs) have shown remarkable potential in image generation, but their sampling efficiency is hindered by the need for numerous denoising steps. Most existing solutions accelerate the sampling process by proposing fast ODE solvers. However, the inevitable discretization errors of the ODE solvers are significantly magnified when the number of function evaluations (NFE) is fewer. In this work, we propose *PFDiff*, a novel *training-free* and *orthogonal* timestep-skipping strategy, which enables existing fast ODE solvers to operate with fewer NFE. Based on two key observations: a significant similarity in the model's outputs at time step size that is not excessively large during the denoising process of existing ODE solvers, and a high resemblance between the denoising process and SGD. PFDiff, by employing gradient replacement from past time steps and foresight updates inspired by Nesterov momentum, rapidly updates intermediate states, thereby reducing unnecessary NFE while correcting for discretization errors inherent in first-order ODE solvers. Experimental results demonstrate that PFDiff exhibits flexible applicability across various pre-trained DPMs, particularly excelling in conditional DPMs and surpassing previous state-of-the-art training-free methods. For instance, using DDIM as a baseline, we achieved 16.46 FID (4 NFE) compared to 138.81 FID with DDIM on ImageNet 64x64 with classifier guidance, and 13.06 FID (10 NFE) on Stable Diffusion with 7.5 guidance scale.

## 1 Introduction

In recent years, Diffusion Probabilistic Models (DPMs) [1–4] have demonstrated exceptional modeling capabilities across various domains including image generation [5–7], video generation [8], text-to-image generation [9, 10], speech synthesis [11], and text-to-3D generation [12, 13]. They have become a key driving force advancing deep generative models. DPMs initiate with a forward process that introduces noise onto images, followed by utilizing a neural network to learn a backward process that incrementally removes noise, thereby generating images [2, 4]. Compared to other generative methods such as Generative Adversarial Networks (GANs) [14] and Variational Autoencoders (VAEs) [15], DPMs not only possess a simpler optimization target but also are capable of producing higher quality samples [5]. However, the generation of high-quality samples via DPMs requires hundreds or thousands of denoising steps, significantly lowering their sampling efficiency and becoming a major barrier to their widespread application.

Existing techniques for rapid sampling in DPMs primarily fall into two categories. First, training-based methods [16–19], which can significantly compress sampling steps, even achieving single-step sampling [19]. However, this compression often comes with a considerable additional training cost, and these methods are challenging to apply to large pre-trained models. Second, training-free samplers [20–30], which typically employ implicit or analytical solutions to Stochastic Differential Equations

**Text Prompts: Winter night with snow -covered rooftops and soft yellow lights.** (Left)
**A Corgi running towards me in Times Square.** (Right)

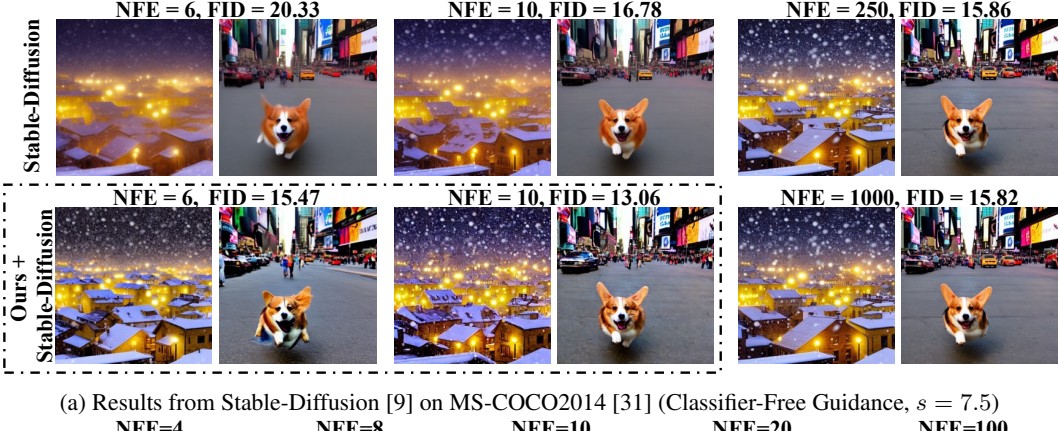

(a) Results from Stable-Diffusion [9] on MS-COCO2014 [31] (Classifier-Free Guidance, $s = 7.5$)

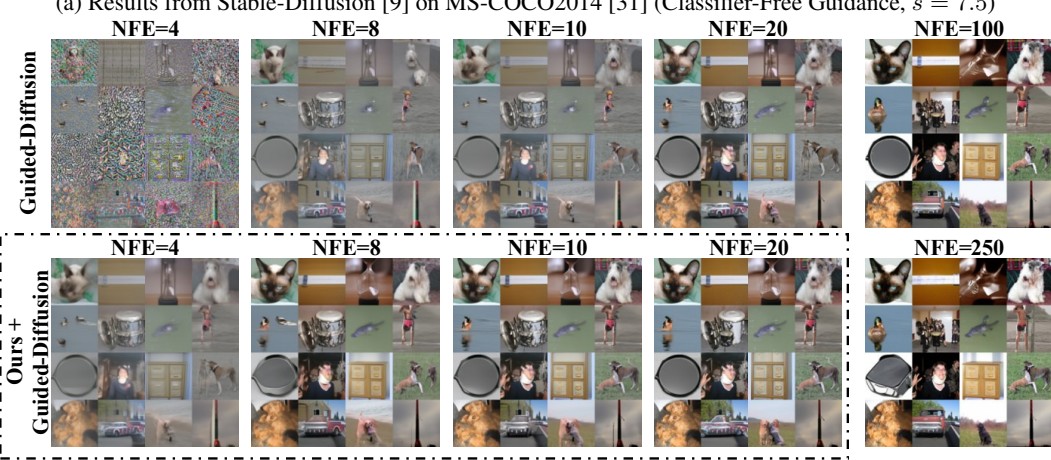

(b) Results from Guided-Diffusion [5] on ImageNet 64x64 [32] (Classifier Guidance, $s = 1.0$)

Figure 1: Sampling by conditional pre-trained DPMs [5, 9] using DDIM [20] and our method PFDiff (dashed box) with DDIM as a baseline, varying the number of function evaluations (NFE).

(SDE)/Ordinary Differential Equations (ODE) for lower-error sampling processes. For instance, Lu et al. [21, 22], by analyzing the semi-linear structure of the ODE solvers for DPMs, have sought to analytically derive optimally the solutions for DPMs' ODE solvers. These training-free sampling strategies can often be used in a plug-and-play fashion, compatible with existing pre-trained DPMs. However, when the NFE is below 10, the discretization error of these training-free methods will be significantly amplified, leading to convergence issues [21, 22], which can still be time-consuming.

To further enhance the sampling speed of DPMs, we have analyzed the potential for improvement in existing training-free accelerated methods. Initially, we observed a notably high similarity in the model's outputs for the existing ODE solvers of DPMs when time step size $\Delta t$ is not extremely large, as illustrated in Fig. 2a. This observation led us to utilize the gradients that have been computed from past time steps to approximate current gradients, thereby reducing unnecessary estimation of noise network. Furthermore, due to the similarities between the sampling process of DPMs and Stochastic Gradient Descent (SGD) [33] as noted in Remark 1, we incorporated a *foresight* update mechanism using Nesterov momentum [34], known for accelerating SGD training. Specifically, we ingeniously employ prior observation to predict future gradients, then utilize the future gradients as a "*springboard*" to facilitate larger update step size $\Delta t$, as shown in Fig. 2b.

Motivated by these insights, we propose *PFDiff*, a timestep-skipping sampling algorithm that rapidly updates the current intermediate state through the gradient guidance of past and future. Notably, PFDiff is *training-free* and *orthogonal* to existing DPMs sampling algorithms, providing a new orthogonal axis for DPMs sampling. Unlike previous orthogonal sampling algorithms that compromise

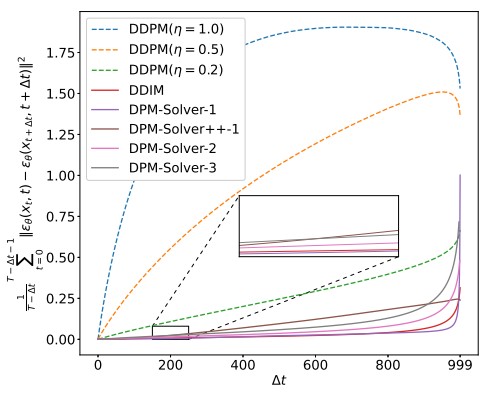
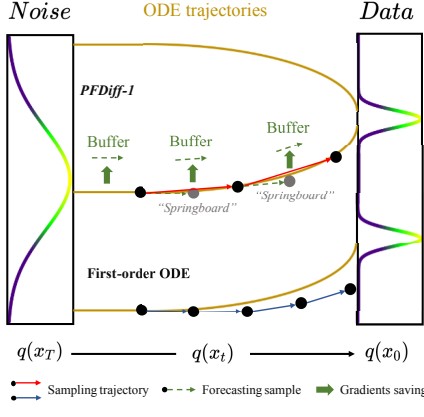

(a) Gradient Changes in SDE/ODE Solvers

(b) Comparison of Sampling Trajectories

Figure 2: (a) The trend of the MSE of the noise network output $\epsilon_\theta(x_t, t)$ over time step size $\Delta t$, where $\eta$ in DDPM [2] comes from $\bar{\sigma}_t$ in Eq. (6). Solid lines: ODE solvers, dashed lines: SDE solvers. (b) Comparison of partial sampling trajectories between PFDiff-1 and a first-order ODE solver, where the update directions are guided by the tangent direction of the sampling trajectories.

sampling quality for speed [28], we prove that PFDiff corrects for errors in the sampling trajectories of first-order ODE solvers. This improves sampling quality while reducing unnecessary NFE in existing ODE solvers, as illustrated in Fig. 2b. To validate the orthogonality and effectiveness of PFDiff, extensive experiments were conducted on both unconditional [2, 4, 20] and conditional [5, 9] pre-trained DPMs, with the visualization experiment of conditional DPMs depicted in Fig. 1. The results indicate that PFDiff significantly enhances the sampling performance of existing ODE solvers. Particularly in conditional DPMs, PFDiff, using only DDIM as the baseline, surpasses the previous state-of-the-art training-free sampling algorithms.

## 2 Background

### 2.1 Diffusion SDEs

Diffusion Probabilistic Models (DPMs) [1–4] aim to generate $D$-dimensional random variables $x_0 \in \mathbb{R}^D$ that follow a data distribution $q(x_0)$. Taking Denoising Diffusion Probabilistic Models (DDPM) [2] as an example, these models introduce noise to the data distribution through a forward process defined over discrete time steps, gradually transforming it into a standard Gaussian distribution $x_T \sim \mathcal{N}(\mathbf{0}, \boldsymbol{I})$. The forward process's latent variables $\{x_t\}_{t \in [0,T]}$ are defined as follows:

$$q(x_t \mid x_0) = \mathcal{N}(x_t \mid \alpha_t x_0, \sigma_t^2 \boldsymbol{I}), \tag{1}$$

where $\alpha_t$ is a scalar function related to the time step $t$, with $\alpha_t^2 + \sigma_t^2 = 1$. In the model's reverse process, DDPM utilizes a neural network model $p_\theta(x_{t-1} \mid x_t)$ to approximate the transition probability $q(x_{t-1} \mid x_t, x_0)$,

$$p_\theta(x_{t-1} \mid x_t) = \mathcal{N}(x_{t-1} \mid \mu_\theta(x_t, t), \sigma_\theta^2(t)\boldsymbol{I}), \tag{2}$$

where $\sigma_\theta^2(t)$ is defined as a scalar function related to the time step $t$. By sampling from a standard Gaussian distribution and utilizing the trained neural network, samples following the data distribution $p_\theta(x_0) = \prod_{t=1}^{T} p_\theta(x_{t-1} \mid x_t)$ can be generated.

Furthermore, Song et al. [4] introduced SDE to model DPMs over continuous time steps, where the forward process is defined as:

$$\mathrm{d}x_t = f(t)x_t\mathrm{d}t + g(t)\mathrm{d}w_t, \quad x_0 \sim q(x_0), \tag{3}$$

where $w_t$ represents a standard Wiener process, and $f$ and $g$ are scalar functions of the time step $t$. It's noteworthy that the forward process in Eq. (1) is a discrete form of Eq. (3), where $f(t) = \frac{\mathrm{d} \log \alpha_t}{\mathrm{d}t}$ and $g^2(t) = \frac{\mathrm{d}\sigma_t^2}{\mathrm{d}t} - 2\frac{\mathrm{d} \log \alpha_t}{\mathrm{d}t}\sigma_t^2$. Song et al. [4] further demonstrated that there exists an equivalent reverse process from time step $T$ to 0 for the forward process in Eq. (3):

$$\mathrm{d}x_t = \left[f(t)x_t - g^2(t)\nabla_x \log q_t(x_t)\right]\mathrm{d}t + g(t)\mathrm{d}\bar{w}_t, \quad x_T \sim q(x_T), \tag{4}$$

where $\bar{w}$ denotes a standard Wiener process. In this reverse process, the only unknown is the *score function* $\nabla_x \log q_t(x_t)$, which can be approximated through neural networks.

## 2.2 Diffusion ODEs

In DPMs based on SDE, the discretization of the sampling process often requires a significant number of time steps to converge, such as the $T = 1000$ time steps used in DDPM [2]. This requirement primarily stems from the randomness introduced at each time step by the SDE. To achieve a more efficient sampling process, Song et al. [4] utilized the Fokker-Planck equation [35] to derive a *probability flow ODE* related to the SDE, which possesses the same marginal distribution at any given time $t$ as the SDE. Specifically, the reverse process ODE derived from Eq. (3) can be expressed as:

$$\mathrm{d}x_t = \left[ f(t)x_t - \frac{1}{2}g^2(t)\nabla_x \log q_t(x_t) \right] \mathrm{d}t, \quad x_T \sim q(x_T). \tag{5}$$

Unlike SDE, ODE avoids the introduction of randomness, thereby allowing convergence to the data distribution in fewer time steps. Song et al. [4] employed a high-order RK45 ODE solver [36], achieving sample quality comparable to SDE at 1000 NFE with only 60 NFE. Furthermore, research such as DDIM [20] and DPM-Solver [21] explored discrete ODE forms capable of converging in fewer NFE. For DDIM, it breaks the Markov chain constraint on the basis of DDPM, deriving a new sampling formula expressed as follows:

$$x_{t-1} = \sqrt{\alpha_{t-1}} \left( \frac{x_t - \sqrt{1 - \alpha_t} \epsilon_\theta(x_t, t)}{\sqrt{\alpha_t}} \right) + \sqrt{1 - \alpha_{t-1} - \bar{\sigma}_t^2} \epsilon_\theta(x_t, t) + \bar{\sigma}_t \epsilon_t, \tag{6}$$

where $\bar{\sigma}_t = \eta \sqrt{(1 - \alpha_{t-1})/(1 - \alpha_t)} \sqrt{1 - \alpha_t/\alpha_{t-1}}$, and $\alpha_t$ corresponds to $\alpha_t^2$ in Eq. (1). When $\eta = 1$, Eq. (6) becomes a form of DDPM [2]; when $\eta = 0$, it degenerates into an ODE, the form adopted by DDIM [20], which can obtain high-quality samples in fewer time steps.

**Remark 1.** *In this paper, we regard the gradient $\mathrm{d}\bar{x}_t$, the noise network output $\epsilon_\theta(x_t, t)$, and the score function $\nabla_x \log q_t(x_t)$ as expressing equivalent concepts. This is because Song et al. [4] demonstrated that $\epsilon_\theta(x_t, t) = -\sigma_t \nabla_x \log q_t(x_t)$. Moreover, we have discovered that any first-order solver of DPMs can be parameterized as $x_{t-1} = \bar{x}_t - \gamma_t \mathrm{d}\bar{x}_t + \xi \epsilon_t$. Taking DDIM [20] as an example, where $\bar{x}_t = \sqrt{\frac{\alpha_{t-1}}{\alpha_t}} x_t$, $\gamma_t = \sqrt{\frac{\alpha_{t-1}}{\alpha_t} - \alpha_{t-1}} - \sqrt{1 - \alpha_{t-1}}$, $\mathrm{d}\bar{x}_t = \epsilon_\theta(x_t, t)$, and $\xi = 0$. This indicates the similarity between SGD and the sampling process of DPMs, a discovery also implicitly suggested in the research of Xue et al. [30] and Wang et al. [37].*

## 3 Method

### 3.1 Solving for reverse process diffusion ODEs

By substituting $\epsilon_\theta(x_t, t) = -\sigma_t \nabla_x \log q_t(x_t)$ [4], Eq. (5) can be rewritten as:

$$\frac{\mathrm{d}x_t}{\mathrm{d}t} = s(\epsilon_\theta(x_t, t), x_t, t) := f(t)x_t + \frac{g^2(t)}{2\sigma_t}\epsilon_\theta(x_t, t), \quad x_T \sim q(x_T). \tag{7}$$

Given an initial value $x_T$, we define the time steps $\{t_i\}_{i=0}^{T}$ to progressively decrease from $t_0 = T$ to $t_T = 0$. Let $\tilde{x}_{t_0} = x_T$ be the initial value. Using $T$ steps of iteration, we compute the sequence $\{\tilde{x}_{t_i}\}_{i=0}^{T}$ to obtain the solution of this ODE. By integrating both sides of Eq. (7), we can obtain the exact solution of this sampling ODE.

$$\tilde{x}_{t_i} = \tilde{x}_{t_{i-1}} + \int_{t_{i-1}}^{t_i} s(\epsilon_\theta(x_t, t), x_t, t)\mathrm{d}t. \tag{8}$$

For any $p$-order ODE solver, Eq. (8) can be discretely represented as:

$$\tilde{x}_{t_{i-1} \to t_i} \approx \tilde{x}_{t_{i-1}} + \sum_{n=0}^{p-1} h(\epsilon_\theta(\tilde{x}_{\hat{t}_n}, \hat{t}_n), \tilde{x}_{\hat{t}_n}, \hat{t}_n) \cdot \Delta\hat{t}, \quad i \in [1, \ldots, T], \tag{9}$$

where $\hat{t}_0 = t_{i-1}$, $\hat{t}_p = t_i$, and $\Delta\hat{t} = \hat{t}_{n+1} - \hat{t}_n$ denote the time step size. The function $h$ represents the different solution methodologies applied by various $p$-order ODE solvers to the function $s$. For the Euler-Maruyama solver [38], $h$ is the identity mapping of $s$. Further, we define

$120$ $\phi(Q, \tilde{x}_{t_{i-1}}, t_{i-1}, t_i) := \tilde{x}_{t_{i-1}} + \sum_{n=0}^{p-1} h(\epsilon_\theta(\tilde{x}_{\hat{t}_n}, \hat{t}_n), \tilde{x}_{\hat{t}_n}, \hat{t}_n) \cdot \Delta\hat{t}$. Here, $\phi$ is any $p$-order ODE

$121$ solver, and buffer $Q = \left( \{\epsilon_\theta(\tilde{x}_{\hat{t}_n}, \hat{t}_n)\}_{n=0}^{p-1}, t_{i-1}, t_i \right)$, where $\hat{t}_0 = t_{i-1}$ and $\hat{t}_p = t_i$.

$122$ When using the ODE solver defined in Eq. (9) for sampling, the choice of $T = 1000$ leads to
$123$ significant inefficiencies in DPMs. The study on DDIM [20] first revealed that by constructing a new
$124$ forward sub-state sequence of length $M + 1$ $(M \leq T)$, $\{\tilde{x}_{t_i}\}_{i=0}^M$, from a subsequence of time steps
$125$ $[0, \ldots, T]$ and reversing this sub-state sequence, it is possible to converge to the data distribution in
$126$ fewer time steps. However, as illustrated in Fig. 2a, for ODE solvers, as the time step $\Delta t = t_i - t_{i-1}$
$127$ increases, the gradient direction changes slowly initially, but undergoes abrupt changes as $\Delta t \to T$.
$128$ This phenomenon indicates that under minimal NFE (i.e., maximal time step size $\Delta t$) conditions, the
$129$ discretization error in Eq. (9) is significantly amplified. Consequently, existing ODE solvers, when
$130$ sampling under minimal NFE, must sacrifice sampling quality to gain speed, making it an extremely
$131$ challenging task to reduce NFE to below 10 [21, 22]. Given this, we aim to develop an efficient
$132$ timestep-skipping sampling algorithm, which reduces NFE while correcting discretization errors,
$133$ thereby ensuring that sampling quality is not compromised, and may even be improved.

$134$ ## 3.2 Sampling guided by past gradients

$135$ For any $p$-order timestep-skipping sampling algorithm for DPMs, the sampling process can be
$136$ reformulated according to Eq. (9) as follows:

$$\tilde{x}_{t_i} \approx \phi(Q, \tilde{x}_{t_{i-1}}, t_{i-1}, t_i), \quad i \in [1, \ldots, M], \tag{10}$$

$137$ where buffer $Q = \left( \{\epsilon_\theta(\tilde{x}_{\hat{t}_n}, \hat{t}_n)\}_{n=0}^{p-1}, t_{i-1}, t_i \right)$ and $[1, \ldots, M]$ is an increasing subsequence of
$138$ $[1, \ldots, T]$. As illustrated in Fig. 2a, when the time step size $\Delta t$ (i.e., $t_i - t_{i-1}$) is not excessively
$139$ large, the MSE of the noise network, defined as $\frac{1}{T - \Delta t} \sum_{t=0}^{T-\Delta t-1} \|\epsilon_\theta(x_t, t) - \epsilon_\theta(x_{t+\Delta t}, t + \Delta t)\|^2$, is
$140$ remarkably similar. This phenomenon is especially pronounced in ODE-based sampling algorithms,
$141$ such as DDIM [20] and DPM-Solver [21]. This observation suggests that there are many unnecessary
$142$ time steps in ODE-based sampling methods during the complete sampling process (e.g., when
$143$ $T = 1000$), which is one of the reasons these methods can generate samples in fewer steps. Based on
$144$ this, we propose replacing the noise network of the current timestep with the output from a previous
$145$ timestep to reduce unnecessary NFE without compromising the quality of the final generated samples.
$146$ Initially, we store the output of the previous timestep's noise network in a *buffer* as follows:

$$Q \xleftarrow{\text{buffer}} \left( \{\epsilon_\theta(\tilde{x}_{\hat{t}_n}, \hat{t}_n)\}_{n=0}^{p-1}, t_{i-1}, t_i \right), \quad \text{where } \hat{t}_0 = t_{i-1}, \hat{t}_p = t_i. \tag{11}$$

$147$ Then, in the current timestep, we directly use the noise network output saved in the buffer from
$148$ the previous timestep to replace the current timestep's noise network output, thereby updating the
$149$ intermediate states to the next timestep, as detailed below:

$$\tilde{x}_{t_{i+1}} \approx \phi(Q, \tilde{x}_{t_i}, t_i, t_{i+1}), \quad \text{where } Q = \left( \{\epsilon_\theta(\tilde{x}_{\hat{t}_n}, \hat{t}_n)\}_{n=0}^{p-1}, t_{i-1}, t_i \right). \tag{12}$$

$150$ By using this approach, we can effectively accelerate the sampling process, reduce unnecessary NFE,
$151$ and ensure the quality of the samples is not affected. The convergence proof is in Appendix B.1.

$152$ ## 3.3 Sampling guided by future gradients

$153$ As stated in Remark 1, considering the similarities between the sampling process of DPMs and SGD
$154$ [33], we introduce a *foresight* update mechanism of Nesterov momentum, utilizing future gradient
$155$ information as a "*springboard*" to assist the current intermediate state in achieving more efficient
$156$ leapfrog updates. Specifically, for the intermediate state $\tilde{x}_{t_{i+1}}$ predicted using past gradients as
$157$ discussed in Sec. 3.2, we first estimate the future gradient and *update* the current *buffer* as follows:

$$Q \xleftarrow{\text{buffer}} \left( \{\epsilon_\theta(\tilde{x}_{\hat{t}_n}, \hat{t}_n)\}_{n=0}^{p-1}, t_{i+1}, t_{i+2} \right), \quad \text{where } \hat{t}_0 = t_{i+1}, \hat{t}_p = t_{i+2}. \tag{13}$$

$158$ Subsequently, leveraging the concept of foresight updates, we predict a further future intermedi-
$159$ ate state $\tilde{x}_{t_{i+2}}$ using the current intermediate state $\tilde{x}_{t_i}$ along with the future gradient information
$160$ corresponding to $\tilde{x}_{t_{i+1}}$, as shown below:

$$\tilde{x}_{t_{i+2}} \approx \phi(Q, \tilde{x}_{t_i}, t_i, t_{i+2}), \quad \text{where } Q = \left( \{\epsilon_\theta(\tilde{x}_{\hat{t}_n}, \hat{t}_n)\}_{n=0}^{p-1}, t_{i+1}, t_{i+2} \right). \tag{14}$$

Furthermore, Zhou et al. [39] performed a Principal Component Analysis (PCA) on the sampling trajectories generated by ODE solvers for DPMs and discovered they almost lie in a two-dimensional plane embedded within a high-dimensional space. This implies that the *Mean Value Theorem* approximately holds during the sampling process using ODE solvers. Specifically, updating the current intermediate state $\tilde{x}_{t_i}$ at an optimal time point $s$ with the corresponding gradient information, *ground truth* $\epsilon_\theta(\tilde{x}_{t_s}, t_s)$, results in the smallest update error, where $s$ is between time points $i$ and $i + 2$. Further, we can reason that for any *first-order* ODE solver, under the same time step, the use of future gradient information $\epsilon_\theta(\tilde{x}_{t_{i+1}}, t_{i+1})$ from Eq. (13) to update the current intermediate state $\tilde{x}_{t_i}$ results in a smaller sampling error compared to using the gradient information at the current time point $\epsilon_\theta(\tilde{x}_{t_i}, t_i)$. A detailed proof is provided in Appendix B.2. However, for higher-order ODE solvers, the solving process implicitly utilizes future gradients as mentioned in Sec. 3.5, and the additional explicit introduction of future gradients increases sampling error. Therefore, when using higher-order ODE solvers as a baseline, the sampling process is accelerated by only using past gradients. It is only necessary to modify Eq. (14) to $\tilde{x}_{t_{i+2}} \approx \phi(Q, \tilde{x}_{t_{i+1}}, t_{i+1}, t_{i+2})$ while keeping $Q$ constant. Ablation experiments can be found in Sec. 4.3.

### 3.4 PFDiff: sampling guided by past and future gradients

Combining Sec. 3.2 and Sec. 3.3, the intermediate state $\tilde{x}_{t_{i+1}}$ obtained through Eq. (12) is used to update the buffer $Q$ in Eq. (13). In this way, we achieve our proposed efficient timestep-skipping algorithm, which we name PFDiff, as shown in Algorithm 1. For higher-order ODE solvers ($p > 1$), PFDiff only utilizes past gradient information, while for first-order ODE solvers ($p = 1$), it uses both past and future gradient information to predict further future intermediate states. Notably, during the iteration from intermediate state $\tilde{x}_{t_i}$ to $\tilde{x}_{t_{i+2}}$, we only perform a single batch computation (NFE = $p$) of the noise network in Eq. (13). Furthermore, we propose that in a single iteration process, $\tilde{x}_{t_{i+2}}$ in Eq. (14) can be modified to $\tilde{x}_{t_{i+(k+1)}}$, achieving a $k$-step skip to sample more distant future intermediate states. Additionally, when $k \neq 1$, the buffer $Q$, which acts as an intermediate "springboard" from Eq. (13), has various computational origins. This can be accomplished by modifying $\tilde{x}_{t_{i+1}}$ in Eq. (12) to $\tilde{x}_{t_{i+l}}$. We collectively refer to this multi-step skipping and different "springboard" selection strategy as PFDiff-$k\_l$ ($l \leq k$). Further algorithmic details can be found in Appendix C. Finally, through the comparison of sampling trajectories between PFDiff-1 and a first-order ODE sampler, as shown in Fig. 2b, PFDiff-1 showcases its capability to correct the sampling trajectory of the first-order ODE sampler while reducing the NFE.

**Proposition 3.1.** *For any given DPM first-order ODE solver $\phi$, the PFDiff-$k\_l$ algorithm can describe the sampling process within an iteration cycle through the following formula:*

$$\tilde{x}_{t_{i+(k+1)}} \approx \phi(\epsilon_\theta(\phi(\epsilon_\theta(\tilde{x}_{t_{i-(k-l+1)}}, t_{i-(k-l+1)}), \tilde{x}_{t_i}, t_i, t_{i+l}), t_{i+l}), \tilde{x}_{t_i}, t_i, t_{i+(k+1)}), \quad (15)$$

---

**Algorithm 1** PFDiff-1

---

**Require:** initial value $x_T$, NFE $N$, model $\epsilon_\theta$, any $p$-order solver $\phi$
1: Define time steps $\{t_i\}_{i=0}^M$ with $M = 2N - 1p$
2: $\tilde{x}_{t_0} \leftarrow x_T$
3: $Q \xleftarrow{\text{buffer}} \left( \{\epsilon_\theta(\tilde{x}_{\hat{t}_n}, \hat{t}_n)\}_{n=0}^{p-1}, t_0, t_1 \right)$, where $\hat{t}_0 = t_0, \hat{t}_p = t_1$         ▷ Initialize buffer
4: $\tilde{x}_{t_1} = \phi(Q, \tilde{x}_{t_0}, t_0, t_1)$
5: **for** $i \leftarrow 1$ to $\frac{M}{p} - 2$ **do**
6:      **if** $(i - 1) \mod 2 = 0$ **then**
7:          $\tilde{x}_{t_{i+1}} = \phi(Q, \tilde{x}_{t_i}, t_i, t_{i+1})$         ▷ Updating guided by past gradients
8:          $Q \xleftarrow{\text{buffer}} \left( \{\epsilon_\theta(\tilde{x}_{\hat{t}_n}, \hat{t}_n)\}_{n=0}^{p-1}, t_{i+1}, t_{i+2} \right)$         ▷ Update buffer (overwrite)
9:          **if** $p = 1$ **then**
10:             $\tilde{x}_{t_{i+2}} = \phi(Q, \tilde{x}_{t_i}, t_i, t_{i+2})$      ▷ Anticipatory updating guided by future gradients
11:          **else if** $p > 1$ **then**
12:             $\tilde{x}_{t_{i+2}} = \phi(Q, \tilde{x}_{t_{i+1}}, t_{i+1}, t_{i+2})$    ▷ The higher-order solver uses only past gradients
13:          **end if**
14:      **end if**
15: **end for**
16: **return** $\tilde{x}_{t_M}$

---

*where the value of $\epsilon_\theta(\tilde{x}_{t_{i-(k-l+1)}}, t_{i-(k-l+1)})$ can be directly obtained from the buffer Q, without the need for additional computations. The iterative process defined by Eq. (15) ensures that the sampling outcomes converge to the data distribution consistent with the solver $\phi$, while effectively correcting errors in the sampling process (Proof in Appendix B).*

It is noteworthy that, although the PFDiff is conceptually orthogonal to the SDE/ODE solvers of DPMs, even when the time size $\Delta t$ is relatively small, the MSE of the noise network in the SDE solver exhibits significant differences, as shown in Fig. 2a. Consequently, PFDiff shows marked improvements on the ODE solver, and our experiments are almost exclusively based on ODE solvers, with exploratory experiments on SDE solvers referred to Sec. 4.1.

### 3.5 Connection with other samplers

**Relationship with $p$-order solver [21, 22, 27].** According to Eq. (10), a single iteration of the $p$-order solver can be represented as:

$$\tilde{x}_{t_{i+1}} \approx \text{Solver} - \text{p}(\left(\{\epsilon_\theta(\tilde{x}_{\hat{t}_n}, \hat{t}_n)\}_{n=0}^{p-1}, t_i, t_{i+1}\right), \tilde{x}_{t_i}, t_i, t_{i+1}), \quad i \in [0, \ldots, M-1]. \quad (16)$$

A single iteration of the $p$-order solver uses $p$ NFE to predict the next intermediate state. The intermediate step gradients obtained during this process can be considered as an approximation of future gradients. This approximation is implicitly contained within the sampling guided by future gradients that we propose. Furthermore, as shown in Eq. (15), a single iteration update of PFDiff based on a first-order solver can be seen as using a 2-order solver with only one NFE.

## 4 Experiments

In this section, we validate the effectiveness of PFDiff as an *orthogonal* and *training-free* sampler through a series of extensive experiments. This sampler can be integrated with any order of ODE solvers, thereby significantly enhancing the sampling efficiency of various types of pre-trained DPMs. To systematically showcase the performance of PFDiff, we categorize the pre-trained DPMs into two main types: conditional and unconditional. Unconditional DPMs are further subdivided into discrete and continuous, while conditional DPMs are subdivided into classifier guidance and classifier-free guidance. In choosing ODE solvers, we utilized the widely recognized first-order DDIM [20], Analytic-DDIM [23], and the higher-order DPM-Solver [21] as baselines. For each experiment, we use the Fréchet Inception Distance (FID↓) [40] as the primary evaluation metric, and provide the experimental results of the Inception Score (IS↑) [41] in the Appendix D.7 for reference. Lastly, apart from the ablation studies on parameters $k$ and $l$ discussed in Sec. 4.3, we showcase the optimal results of PFDiff-$k\_l$ (where $k = 1, 2, 3$ and $l \leq k$) across six configurations as a performance demonstration of PFDiff. As described in Appendix C, this does not increase the computational burden in practical applications. All experiments were conducted on an NVIDIA RTX 3090 GPU.

### 4.1 Unconditional sampling

For unconditional DPMs, we selected discrete DDPM [2] and DDIM [20], as well as pre-trained models from continuous ScoreSDE [4], to assess the effectiveness of PFDiff. For these pre-trained models, all experiments sampled 50k instances to compute evaluation metrics.

For unconditional discrete DPMs, we first select first-order ODE solvers DDIM [20] and Analytic-DDIM [23] as baselines, while implementing SDE-based DDPM [2] and Analytic-DDPM [23] methods for comparison, where $\eta = 1.0$ is from $\bar{\sigma}_t$ in Eq. (6). We conduct experiments on the CIFAR10 [42] and CelebA 64x64 [43] datasets using the quadratic time steps employed by DDIM. By varying the NFE from 6 to 20, the evaluation metric FID↓ is shown in Figs. 3a and 3b. Additionally, experiments with uniform time steps are conducted on the CelebA 64x64, LSUN-bedroom 256x256 [44], and LSUN-church 256x256 [44] datasets, with more results available in Appendix D.2. Our experimental results demonstrate that PFDiff, based on pre-trained models of discrete unconditional DPMs, significantly improves the sampling efficiency of DDIM and Analytic-DDIM samplers across multiple datasets. For instance, on the CIFAR10 dataset, PFDiff combined with DDIM achieves a FID of 4.10 with only 15 NFE, comparable to DDIM's performance of 4.04 FID with 1000 NFE. This is something other time-step skipping algorithms [23, 28] that sacrifice sampling quality for speed

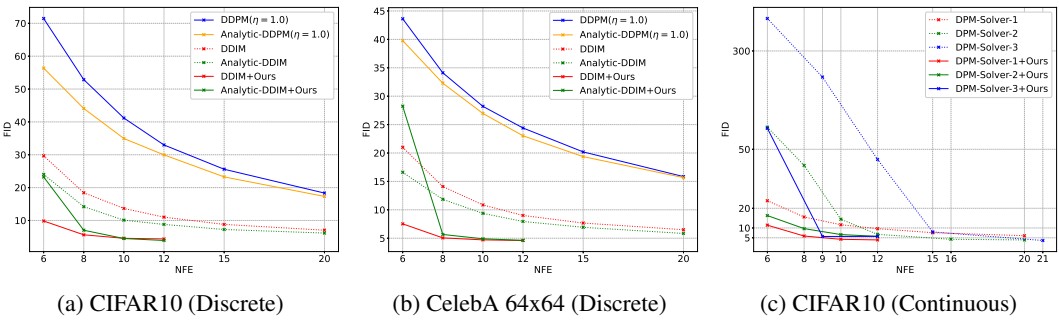

(a) CIFAR10 (Discrete)  (b) CelebA 64x64 (Discrete)  (c) CIFAR10 (Continuous)

Figure 3: Unconditional sampling results. We report the FID↓ for different methods by varying the number of function evaluations (NFE), evaluated on 50k samples.

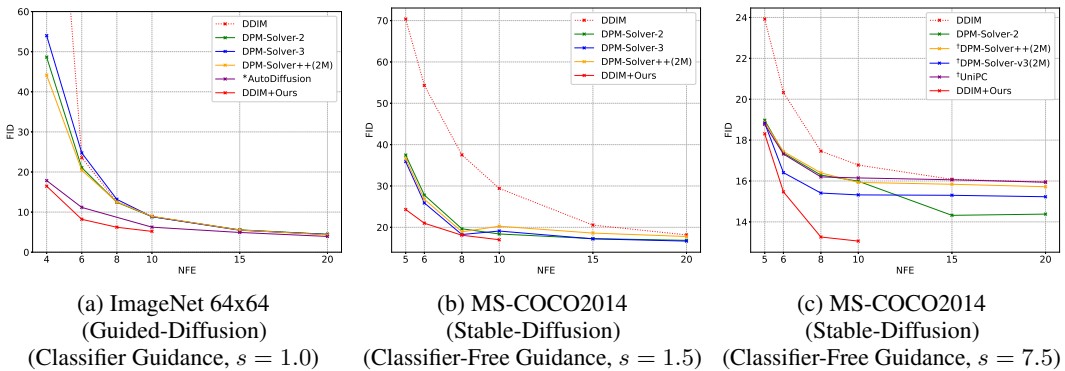

(a) ImageNet 64x64
(Guided-Diffusion)
(Classifier Guidance, $s = 1.0$)

(b) MS-COCO2014
(Stable-Diffusion)
(Classifier-Free Guidance, $s = 1.5$)

(c) MS-COCO2014
(Stable-Diffusion)
(Classifier-Free Guidance, $s = 7.5$)

Figure 4: Conditional sampling results. We report the FID↓ for different methods by varying the NFE. Evaluated: ImageNet 64x64 with 50k, others with 10k samples. *AutoDiffusion [26] method requires additional search costs. †We borrow the results reported in DPM-Solver-v3 [27] directly.

cannot achieve. Furthermore, in Appendix D.2, by varying $\eta$ from 1.0 to 0.0 in Eq. (6) to control the scale of noise introduced by SDE, we observe that as $\eta$ decreases (reducing noise introduction), the performance of PFDiff gradually improves. This once again validates our assumption proposed in Sec. 3.2, based on Fig. 2a, that there is a significant similarity in the model's outputs at the time step size that is not excessively large for the existing ODE solvers.

For unconditional continuous DPMs, we choose the DPM-Solver-1, -2 and -3 [21] as the baseline to verify the effectiveness of PFDiff as an orthogonal timestep-skipping algorithm on the first and higher-order ODE solvers. We conducted experiments on the CIFAR10 [42] using quadratic time steps, varying the NFE. The experimental results using FID↓ as the evaluation metric are shown in Fig. 3c. More experimental details can be found in Appendix D.3. We observe that PFDiff consistently improves the sampling performance over the baseline with fewer NFE settings, particularly in cases where higher-order ODE solvers fail to converge with a small NFE (below 10) [21].

## 4.2 Conditional sampling

For conditional DPMs, we selected the pre-trained models of the widely recognized classifier guidance paradigm, ADM-G [5], and the classifier-free guidance paradigm, Stable-Diffusion [9], to validate the effectiveness of PFDiff. We employed uniform time steps setting and the DDIM [20] ODE solver as a baseline across all datasets. Evaluation metrics were computed by sampling 50k samples on the ImageNet 64x64 [32] dataset for ADM-G and 10k samples on other datasets, including ImageNet 256x256 [32] in ADM-G and MS-COCO2014 [31] in Stable-Diffusion.

For conditional DPMs employing the classifier guidance paradigm, we conducted experiments on the ImageNet 64x64 dataset [32] with a guidance scale ($s$) set to 1.0. For comparison, we implemented DPM-Solver-2 and -3 [21], and DPM-Solver++(2M) [22], which exhibit the best performance on conditional DPMs. Additionally, we introduced the AutoDiffusion method [26] using DDIM as a

baseline for comparison, noting that this method incurs additional search costs. We compared FID↓ scores by varying the NFE as depicted in Fig. 4a, with corresponding visual comparisons shown in Fig. 1b. We observed that PFDiff reduced the FID from 138.81 with 4 NFE in DDIM to 16.46, achieving an 88.14% improvement in quality. The visual results in Fig. 1b further demonstrate that, at the same NFE setting, PFDiff achieves higher-quality sampling. Furthermore, we evaluated PFDiff's sampling performance based on DDIM on the large-scale ImageNet 256x256 dataset [32]. Detailed results are provided in Appendix D.4.

For conditional, classifier-free guidance paradigms of DPMs, we employed the `sd-v1-4` checkpoint and computed the FID↓ scores on the validation set of MS-COCO2014 [31]. We conducted experiments with a guidance scale ($s$) set to 7.5 and 1.5. For comparison, we implemented DPM-Solver-2 and -3 [21], and DPM-Solver++(2M) [22] methods. At $s = 7.5$, we introduced the state-of-the-art method reported in DPM-Solver-v3 [27] for comparison, along with DPM-Solver++(2M) [22], UniPC [29], and DPM-Solver-v3(2M) [27]. The FID↓ metrics by varying the NFE are presented in Figs. 4b and 4c, with additional visual results illustrated in Fig. 1a. We observed that PFDiff, solely based on DDIM, achieved state-of-the-art results during the sampling process of Stable-Diffusion, thus demonstrating the efficacy of PFDiff. Further experimental details can be found in Appendix D.5.

### 4.3 Ablation study

We conducted ablation experiments on the six different algorithm configurations of PFDiff mentioned in Appendix C, with $k = 1, 2, 3$ ($l \leq k$). Specifically, we evaluated the FID↓ scores on the unconditional and conditional pre-trained DPMs [2, 4, 5, 9] by varying the NFE. Detailed experimental setups and results can be found in Appendix D.6.1. The experimental results indicate that for various pre-trained DPMs, the choice of parameters $k$ and $l$ is not critical, as most combinations of $k$ and $l$ within PFDiff can enhance the sampling efficiency over the baseline. Moreover, with $k = 1$ fixed, PFDiff-1 can significantly improve the baseline's sampling quality within the range of 8∼20 NFE. For even better sampling quality, one can sample a small subset of examples (e.g., 5k) to compute evaluation metrics or directly conduct visual analysis, easily identifying the most effective $k$ and $l$ combinations.

To validate the PFDiff algorithm as mentioned in Sec. 3.3, which necessitates the joint guidance of past and future gradients for first-order ODE solvers, and only past gradients for higher-order ODE solvers, offering a more effective means of accelerating baseline sampling. This study employs the first-order ODE solver DDIM [20] as the baseline, isolating the effects of both past and future gradients, and uses the higher-order ODE solver DPM-Solver [21] as the baseline, removing the influence of future gradients for ablation experiments. Specific experimental configurations and results are shown in Appendix D.6.2. The results indicate that, as described by the PFDiff algorithm in Sec. 3.3, it is possible to further enhance the sampling efficiency of ODE solvers of any order.

## 5 Conclusion

In this paper, based on the recognition that the ODE solvers of DPMs exhibit significant similarity in model outputs when the time step size is not excessively large, and with the aid of a foresight update mechanism, we propose PFDiff, a novel method that leverages the gradient guidance from both past and future to rapidly update the current intermediate state. This approach effectively reduces the unnecessary number of function evaluations (NFE) in the ODE solvers and significantly corrects the errors of first-order ODE solvers during the sampling process. Extensive experiments demonstrate the orthogonality and efficacy of PFDiff on both unconditional and conditional pre-trained DPMs, especially in conditional pre-trained DPMs where PFDiff outperforms previous state-of-the-art training-free sampling methods.

**Limitations and broader impact**  Although PFDiff can effectively accelerate the sampling speed of existing ODE solvers, it still lags behind the sampling speed of training-based acceleration methods and one-step generation paradigms such as GANs. Moreover, there is no universal setting for the optimal combination of parameters $k$ and $l$ in PFDiff; adjustments are required according to different pre-trained DPMs and NFE. It is noteworthy that PFDiff may be utilized to accelerate the generation of malicious content, thereby having a detrimental impact on society.

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

## A  Related work

While the solvers for Diffusion Probabilistic Models (DPMs) are categorized into two types, SDE and ODE, most current accelerated sampling techniques are based on ODE solvers due to the observation that the stochastic noise introduced by SDE solvers hampers rapid convergence. ODE-based solvers are further divided into training-based methods [16–19] and training-free samplers [20–30]. Training-based methods can notably reduce the number of sampling steps required for DPMs. An example of such a method is the knowledge distillation algorithm proposed by Song et al. [19], which achieves one-step sampling for DPMs. This sampling speed is comparable to that of GANs [14] and VAEs [15]. However, these methods often entail significant additional costs for distillation training. This requirement poses a challenge when applying them to large pre-trained DPM models. Therefore, our work primarily focuses on training-free, ODE-based accelerated sampling strategies.

Training-free accelerated sampling techniques based on ODE can generally be applied in a plug-and-play manner, adapting to existing pre-trained DPMs. These methods can be categorized based on the order of the ODE solver—that is, the NFE required per sampling iteration—into first-order [20, 23–25] and higher-order [21, 22, 27, 29, 36]. Typically, higher-order ODE solvers tend to sample at a faster rate, but may fail to converge when the NFE is low (below 10), sometimes performing even worse than first-order ODE solvers. In addition, there are orthogonal techniques for accelerated sampling. For instance, Li et al. [26] build upon existing ODE solvers and use search algorithms to find optimal sampling sub-sequences and model structures to further speed up the sampling process; Ma et al. [28] observe that the low-level features of noise networks at adjacent time steps exhibit similarities, and they use caching techniques to substitute some of the network's low-level features, thereby further reducing the number of required time steps.

The algorithm we propose belongs to the class of training-free and orthogonal accelerated sampling techniques, capable of further accelerating the sampling process on the basis of existing first-order and higher-order ODE solvers. Compared to the aforementioned orthogonal sampling techniques, even though the skipping strategy proposed by Ma et al. [28] effectively accelerates the sampling process, it may do so at the cost of reduced sampling quality, making it challenging to reduce the NFE below 50. Although Li et al. [26] can identify more optimal subsampling sequences and model structures, this implies higher search costs. In contrast, our proposed orthogonal sampling algorithm is more efficient in skipping time steps. First, our skipping strategy does not require extensive search costs. Second, we can correct the sampling errors of first-order ODE solvers while reducing the number of sampling steps required by existing ODE solvers, achieving more efficient accelerated sampling.

## B  Proof of convergence and error correction for PFDiff

In this section, we prove the convergence of PFDiff and elaborate on how it theoretically corrects first-order ODE solver errors. To delve deeper into PFDiff, we propose the following assumptions:

**Assumption B.1.** *When the time step size $\Delta t = t_i - t_{i-(k-l+1)}$ is not excessively large, the output estimates of the noise network based on the $p$-order ODE solver at different time steps are approximately the same, that is, $\left( \{\epsilon_\theta(\tilde{x}_{\hat{t}_n}, \hat{t}_n)\}_{n=0}^{p-1}, t_i, t_{i+l} \right) \approx \left( \{\epsilon_\theta(\tilde{x}_{\hat{t}_n}, \hat{t}_n)\}_{n=0}^{p-1}, t_{i-(k-l+1)}, t_i \right)$.*

**Assumption B.2.** *For the integral from time step $t_i$ to $t_{i+(k+1)}$, $\int_{t_i}^{t_{i+(k+1)}} s(\epsilon_\theta(x_t, t), x_t, t)\mathrm{d}t$, there exist intermediate time steps $t_{\tilde{s}}, t_s \in (t_i, t_{i+(k+1)})$ such that $\int_{t_i}^{t_{i+(k+1)}} s(\epsilon_\theta(x_t, t), x_t, t)\mathrm{d}t = s(\epsilon_\theta(x_{t_{\tilde{s}}}, t_{\tilde{s}}), x_{t_{\tilde{s}}}, t_{\tilde{s}}) \cdot (t_{i+(k+1)} - t_i) = h(\epsilon_\theta(x_{t_s}, t_s), x_{t_s}, t_s) \cdot (t_{i+(k+1)} - t_i)$ holds, where the definition of the function $h$ remains consistent with Sec. 3.1.*

The first assumption is based on the observation in Fig. 2a that when $\Delta t$ is not excessively large, the MSE of the noise network remains almost unchanged across different time steps. The second assumption is based on the *Mean Value Theorem* for Integrals, which states that if $f(x)$ is a continuous real-valued function on a closed interval $[a, b]$, then there exists at least one point $c \in [a, b]$ such that $\int_a^b f(x)\mathrm{d}x = f(c)(b - a)$ holds. It is important to note that the Mean Value Theorem for Integrals originally applies to real-valued functions and does not directly apply to vector-valued functions [45]. However, the study by Zhou et al. [39] using Principal Component Analysis (PCA) on the trajectories of the ODE solvers for DPMs demonstrates that these trajectories are primarily distributed

on a two-dimensional plane, which allows the Mean Value Theorem for Integrals to approximately hold under Assumption B.2.

## B.1 Proof of convergence for sampling guided by past gradients

Starting from Eq. (8), we consider an iteration process of a $p$-order ODE solver from $\tilde{x}_{t_i}$ to $\tilde{x}_{t_{i+l}}$, where $l$ is the "*springboard*" choice determined by PFDiff-$k\_l$. This iterative process can be expressed as:

$$\tilde{x}_{t_{i+l}} = \tilde{x}_{t_i} + \int_{t_i}^{t_{i+l}} s(\epsilon_\theta(x_t, t), x_t, t) \mathrm{d}t. \tag{B.1}$$

Discretizing Eq. (B.1) yields:

$$\tilde{x}_{t_i \to t_{i+l}} \approx \tilde{x}_{t_i} + \sum_{n=0}^{p-1} h(\epsilon_\theta(\tilde{x}_{\hat{t}_n}, \hat{t}_n), \tilde{x}_{\hat{t}_n}, \hat{t}_n) \cdot (\hat{t}_{n+1} - \hat{t}_n), \tag{B.2}$$

where $\hat{t}_0 = t_i$ and $\hat{t}_p = t_{i+l}$. Consistent with Sec. 3.1, the function $h$ represents the different solution methodologies applied by various $p$-order ODE solvers to the function $s$. To accelerate sampler convergence and reduce unnecessary evaluations of the NFE, we adopt Assumption B.1, namely guiding the sampling of the current intermediate state by utilizing past gradient information. Specifically, we approximate that $\left( \{\epsilon_\theta(\tilde{x}_{\hat{t}_n}, \hat{t}_n)\}_{n=0}^{p-1}, t_i, t_{i+l} \right) \approx \left( \{\epsilon_\theta(\tilde{x}_{\hat{t}_n}, \hat{t}_n)\}_{n=0}^{p-1}, t_{i-(k-l+1)}, t_i \right)$, where $k$ represents the number of steps skipped in one iteration by PFDiff-$k\_l$. Eq. (B.2) can be rewritten as:

$$\begin{aligned}
\tilde{x}_{t_i \to t_{i+l}} &\approx \tilde{x}_{t_i} + \sum_{n=i}^{i+l-1} h(\epsilon_\theta(\tilde{x}_{t_n}, t_n), \tilde{x}_{t_n}, t_n) \cdot (t_{n+1} - t_n) \\
&\approx \tilde{x}_{t_i} + \sum_{n=i-(k-l+1)}^{i-1} h(\epsilon_\theta(\tilde{x}_{t_n}, t_n), \tilde{x}_{t_n}, t_n) \cdot (t_{n+1} - t_n) \\
&= \phi\left( \left( \{\epsilon_\theta(\tilde{x}_{\hat{t}_n}, \hat{t}_n)\}_{n=0}^{p-1}, t_{i-(k-l+1)}, t_i \right), \tilde{x}_{t_i}, t_i, t_{i+l} \right),
\end{aligned} \tag{B.3}$$

where $\phi$ is any $p$-order ODE solver. Eq. (B.3) demonstrates that under Assumption B.1, for any $p$-order ODE solver $\phi$, PFDiff-$k\_l$ utilizes past gradient information as a substitute for current gradient information to update the current intermediate state. This method not only reduces the NFE but also approximates the solution of $\tilde{x}_{t_{i+l}}$, ensuring convergence to the data distribution corresponding to the solver $\phi$. It is important to note that the sampling process described in Eq. (B.3) relies solely on past gradient information and does not estimate the output of the noise network based on the current intermediate state.

In particular, within Proposition 3.1 for any first-order ($p = 1$) ODE solver $\phi$, according to Eq. (B.3), we can approximate $\tilde{x}_{t_{i+l}} \approx \phi(\epsilon_\theta(\tilde{x}_{t_{i-(k-l+1)}}, t_{i-(k-l+1)}), \tilde{x}_{t_i}, t_i, t_{i+l})$. Thus, Eq. (15) can be rewritten as:

$$\tilde{x}_{t_{i+(k+1)}} \approx \phi(\epsilon_\theta(\tilde{x}_{t_{i+l}}, t_{i+l}), \tilde{x}_{t_i}, t_i, t_{i+(k+1)}). \tag{B.4}$$

For any first-order ODE solver $\phi$, Eq. (B.3) and (B.4) utilize the gradient information from both past and future to constitute a complete sampling iteration process for PFDiff-$k\_l$. Eq. (B.4) indicates that under the Assumption B.1 and upon the convergence of Eq. (B.4), PFDiff-$k\_l$ is guaranteed to converge to the data distribution corresponding to the sampler $\phi$ for any first-order ODE solver.

## B.2 Error correction and proof of convergence of Proposition 3.1

Based on Eq. (8), we consider an iteration process of a first-order ($p = 1$) ODE solver from $\tilde{x}_{t_i}$ to $\tilde{x}_{t_{i+(k+1)}}$, which can be expressed as:

$$\begin{aligned}
\tilde{x}_{t_{i+(k+1)}} &= \tilde{x}_{t_i} + \int_{t_i}^{t_{i+(k+1)}} s(\epsilon_\theta(x_t, t), x_t, t) \mathrm{d}t \\
&\approx \tilde{x}_{t_i} + h(\epsilon_\theta(\tilde{x}_{t_i}, t_i), \tilde{x}_{t_i}, t_i) \cdot (t_{i+(k+1)} - t_i) \\
&= \phi(\epsilon_\theta(\tilde{x}_{t_i}, t_i), \tilde{x}_{t_i}, t_i, t_{i+(k+1)}),
\end{aligned} \tag{B.5}$$

507 where the second line of Eq. (B.5) is obtained by discretizing the first line with an existing first-order
508 ODE solver ($p = 1$), and the definition of $\phi$ and $h$ are consistent with Appendix B.1. It is well-known
509 that the discretization method used in Eq. (B.5) restricts the sampling step size $\Delta t = t_{i+(k+1)} - t_i$
510 of the first-order ODE solver. A too-large step size will cause the first-order ODE solver to not
511 converge. This indicates that although Assumption B.2 points out that the sampling trajectory of
512 the first-order ODE solver lies on a two-dimensional plane, this trajectory is not a straight line (if
513 it were a straight line, a larger sampling step size could be used). Therefore, using $\epsilon_\theta(\tilde{x}_{t_i}, t_i)$ for
514 discretized sampling in Eq. (B.5) introduces a significant sampling error, as shown by the first-order
515 ODE sampling trajectory in Fig. 2b. To reduce the first-order ODE solver sampling error, we have
516 revised Eq. (B.5) based on Assumption B.2, as follows:

$$
\begin{aligned}
\tilde{x}_{t_{i+(k+1)}} &= \tilde{x}_{t_i} + \int_{t_i}^{t_{i+(k+1)}} s(\epsilon_\theta(x_t, t), x_t, t)\mathrm{d}t \\
&= \tilde{x}_{t_i} + s(\epsilon_\theta(\tilde{x}_{t_{\tilde{s}}}, t_{\tilde{s}}), \tilde{x}_{t_{\tilde{s}}}, t_{\tilde{s}}) \cdot (t_{i+(k+1)} - t_i) \\
&= \tilde{x}_{t_i} + h(\epsilon_\theta(\tilde{x}_{t_s}, t_s), \tilde{x}_{t_s}, t_s) \cdot (t_{i+(k+1)} - t_i) \\
&= \phi(\epsilon_\theta(\tilde{x}_{t_s}, t_s), \tilde{x}_{t_i}, t_i, t_{i+(k+1)}) \\
&\approx \phi(\epsilon_\theta(\tilde{x}_{t_{i+l}}, t_{i+l}), \tilde{x}_{t_i}, t_i, t_{i+(k+1)}),
\end{aligned}
\tag{B.6}
$$

517 where $k$ and $l$ are determined by the selected PFDiff-$k\_l$ and the second and third lines are obtained
518 based on Assumption B.2. Combining Eq. (B.6) and Eq. (B.3) leads to the complete Eq. (15),
519 thereby completing the convergence proof of Proposition 3.1. Moreover, $t_s$ falls within the interval
520 $[t_i, t_{i+(k+1)}]$, and since the sampling trajectory of the first-order ODE solver is not a straight line,
521 generally $t_s \neq t_i$ and $t_s \neq t_{i+(k+1)}$. The interval $[t_i, t_{i+(k+1)}]$ is amended to $(t_i, t_{i+(k+1)})$. By
522 adopting the *foresight* update mechanism of the Nesterov momentum [34], and guiding the current in-
523 termediate state sampling with future gradient information, we replace $\epsilon_\theta(\tilde{x}_{t_s}, t_s)$ with $\epsilon_\theta(\tilde{x}_{t_{i+l}}, t_{i+l})$.
524 According to the definition of PFDiff-$k\_l$, $t_{i+l}$ also lies within the interval $(t_i, t_{i+(k+1)})$, and for
525 different combinations of $k$ and $l$, this means searching and approximating the *ground truth* $t_s$ within
526 the interval $(t_i, t_{i+(k+1)})$. Among the six different versions of PFDiff-$k\_l$ defined in Appendix C, we
527 believe that the optimal $t_s$ has been approximated. Compared to the direct discretization of $\epsilon_\theta(x_t, t)$
528 in Eq. (B.5), we corrected the sampling error of the first-order ODE solver and proved its convergence
529 by guiding sampling based on the future gradient information $\epsilon_\theta(\tilde{x}_{t_{i+l}}, t_{i+l})$ under Assumption B.2,
530 as shown in the sampling trajectory of PFDiff-1 in Fig. 2b. Together with this section and Appendix
531 B.1, this completes the error correction and convergence proof of Proposition 3.1.

## C  Algorithms of PFDiffs

533 As described in Sec. 3.4, during a single iteration, we can leverage the *foresight* update mechanism to
534 skip to a more distant future. Specifically, we modify Eq. (14) to $\tilde{x}_{t_{i+(k+1)}} \approx \phi(Q, \tilde{x}_{t_i}, t_i, t_{i+(k+1)})$
535 to achieve a $k$-step skip. We refer to this method as PFDiff-$k$. Additionally, when $k \neq 1$, the
536 computation of the buffer $Q$, originating from Eq. (13), presents different selection choices. We
537 modify Eq. (12) to $\tilde{x}_{t_{i+l}} \approx \phi(Q, \tilde{x}_{t_i}, t_i, t_{i+l}), l \leq k$ to denote different "*springboard*" choices with
538 the parameter $l$. This strategy of multi-step skips and varying "springboard" choices is collectively
539 termed as PFDiff-$k\_l$ ($l \leq k$). Consequently, based on modifications to parameters $k$ and $l$ in Eq.
540 (12) and Eq. (14), Eq. (13) is updated to $Q \xleftarrow{\text{buffer}} \left( \{\epsilon_\theta(\tilde{x}_{\hat{t}_n}, \hat{t}_n)\}_{n=0}^{p-1}, t_{i+l}, t_{i+(k+1)} \right)$, and Eq. (11)
541 is updated to $Q \xleftarrow{\text{buffer}} \left( \{\epsilon_\theta(\tilde{x}_{\hat{t}_n}, \hat{t}_n)\}_{n=0}^{p-1}, t_{i-(k-l+1)}, t_i \right)$. When $k = 1$, since $l \leq k$, then $l = 1$,
542 and PFDiff-$k\_l$ is the same as PFDiff-1, as shown in Algorithm 1 in Sec. 3.4. When $k = 2$, $l$
543 can be either 1 or 2, forming Algorithms PFDiff-2_1 and PFDiff-2_2, as shown in Algorithm 2.
544 Furthermore, when $k = 3$, this forms three different versions of PFDiff-3, as shown in Algorithm
545 3. In this study, we utilize the optimal results from the six configurations of PFDiff-$k\_l$ ($k = 1, 2, 3$
546 ($l \leq k$)) to demonstrate the performance of PFDiff. As described in Appendix B.2, this is essentially
547 an approximation of the *ground truth* $t_s$. Through these six different algorithm configurations, we
548 approximately search for the optimal $t_s$. It is important to note that despite using six different
549 algorithm configurations, this does not increase the computational burden in practical applications.
550 This is because, by visual analysis of a small number of generated images or computing specific
551 evaluation metrics, one can effectively select the algorithm configuration with the best performance.
552 Moreover, even without any selection, directly using the PFDiff-1 configuration can achieve significant

performance improvements on top of existing ODE solvers, as shown in the ablation study results in Sec. 4.3.

---

**Algorithm 2** PFDiff-2

---

**Require:** initial value $x_T$, NFE $N$, model $\epsilon_\theta$, any $p$-order solver $\phi$, skip type $l$
 1: Define time steps $\{t_i\}_{i=0}^M$ with $M = 3N - 2p$
 2: $\tilde{x}_{t_0} \leftarrow x_T$
 3: $Q \xleftarrow{\text{buffer}} \left( \left\{ \epsilon_\theta(\tilde{x}_{\hat{t}_n}, \hat{t}_n) \right\}_{n=0}^{p-1}, t_0, t_1 \right)$, where $\hat{t}_0 = t_0, \hat{t}_p = t_1$        ▷ Initialize buffer
 4: $\tilde{x}_{t_1} = \phi(Q, \tilde{x}_{t_0}, t_0, t_1)$
 5: **for** $i \leftarrow 1$ to $\frac{M}{p} - 3$ **do**
 6:     **if** $(i - 1) \bmod 3 = 0$ **then**
 7:       **if** $l = 1$ **then**                         ▷ PFDiff-2_1
 8:         $\tilde{x}_{t_{i+1}} = \phi(Q, \tilde{x}_{t_i}, t_i, t_{i+1})$       ▷ Updating guided by past gradients
 9:         $Q \xleftarrow{\text{buffer}} \left( \left\{ \epsilon_\theta(\tilde{x}_{\hat{t}_n}, \hat{t}_n) \right\}_{n=0}^{p-1}, t_{i+1}, t_{i+3} \right)$      ▷ Update buffer (overwrite)
10:       **else if** $l = 2$ **then**                 ▷ PFDiff-2_2
11:         $\tilde{x}_{t_{i+2}} = \phi(Q, \tilde{x}_{t_i}, t_i, t_{i+2})$       ▷ Updating guided by past gradients
12:         $Q \xleftarrow{\text{buffer}} \left( \left\{ \epsilon_\theta(\tilde{x}_{\hat{t}_n}, \hat{t}_n) \right\}_{n=0}^{p-1}, t_{i+2}, t_{i+3} \right)$      ▷ Update buffer (overwrite)
13:       **end if**
14:       **if** $p = 1$ **then**
15:         $\tilde{x}_{t_{i+3}} = \phi(Q, \tilde{x}_{t_i}, t_i, t_{i+3})$     ▷ Anticipatory updating guided by future gradients
16:       **else if** $p > 1$ **then**
17:         $\tilde{x}_{t_{i+3}} = \phi(Q, \tilde{x}_{t_{i+l}}, t_{i+l}, t_{i+3})$    ▷ The higher-order solver uses only past gradients
18:       **end if**
19:     **end if**
20: **end for**
21: **return** $\tilde{x}_{t_M}$

---

**Algorithm 3** PFDiff-3

---

**Require:** initial value $x_T$, NFE $N$, model $\epsilon_\theta$, any $p$-order solver $\phi$, skip type $l$
 1: Define time steps $\{t_i\}_{i=0}^M$ with $M = 4N - 3p$
 2: $\tilde{x}_{t_0} \leftarrow x_T$
 3: $Q \xleftarrow{\text{buffer}} \left( \left\{ \epsilon_\theta(\tilde{x}_{\hat{t}_n}, \hat{t}_n) \right\}_{n=0}^{p-1}, t_0, t_1 \right)$, where $\hat{t}_0 = t_0, \hat{t}_p = t_1$        ▷ Initialize buffer
 4: $\tilde{x}_{t_1} = \phi(Q, \tilde{x}_{t_0}, t_0, t_1)$
 5: **for** $i \leftarrow 1$ to $\frac{M}{p} - 4$ **do**
 6:     **if** $(i - 1) \bmod 4 = 0$ **then**
 7:       $\tilde{x}_{t_{i+4}} = \phi(Q, \tilde{x}_{t_i}, t_i, t_{i+l})$         ▷ Updating guided by past gradients
 8:       $Q \xleftarrow{\text{buffer}} \left( \left\{ \epsilon_\theta(\tilde{x}_{\hat{t}_n}, \hat{t}_n) \right\}_{n=0}^{p-1}, t_{i+l}, t_{i+4} \right)$      ▷ Update buffer (overwrite)
 9:       **if** $p = 1$ **then**
10:         $\tilde{x}_{t_{i+4}} = \phi(Q, \tilde{x}_{t_i}, t_i, t_{i+4})$     ▷ Anticipatory updating guided by future gradients
11:       **else if** $p > 1$ **then**
12:         $\tilde{x}_{t_{i+4}} = \phi(Q, \tilde{x}_{t_{i+l}}, t_{i+l}, t_{i+4})$    ▷ The higher-order solver uses only past gradients
13:       **end if**
14:     **end if**
15: **end for**
16: **return** $\tilde{x}_{t_M}$

---

# D    Additional experiment results

In this section, we provide further supplements to the experiments on both unconditional and conditional pre-trained Diffusion Probabilistic Models (DPMs) as mentioned in Sec. 4. Through these additional supplementary experiments, we more fully validate the effectiveness of PFDiff as an

orthogonal and training-free sampler. Unless otherwise stated, the selection of pre-trained DPMs, choice of baselines, algorithm configurations, GPU utilization, and other related aspects in this section are consistent with those described in Sec. 4.

## D.1 License

In this section, we list the used datasets, codes, and their licenses in Table 1.

Table 1: The used datasets, codes, and their licenses.

| Name | URL | License |
|------|-----|---------|
| CIFAR10 [42] | https://www.cs.toronto.edu/∼kriz/cifar.html | \ |
| CelebA 64x64 [43] | https://mmlab.ie.cuhk.edu.hk/projects/CelebA.html | \ |
| LSUN-Bedroom [44] | https://www.yf.io/p/lsun | \ |
| LSUN-Church [44] | https://www.yf.io/p/lsun | \ |
| ImageNet [32] | https://image-net.org/ | \ |
| MS-COCO2014 [31] | https://cocodataset.org/ | CC BY 4.0 |
| ScoreSDE [4] | https://github.com/yang-song/score_sde_pytorch | Apache-2.0 |
| DDIM [20] | https://github.com/ermongroup/ddim/tree/main | MIT |
| Analytic-DPM [23] | https://github.com/baofff/Analytic-DPM | \ |
| DPM-Solver [21] | https://github.com/LuChengTHU/dpm-solver | MIT |
| DPM-Solver++ [22] | https://github.com/LuChengTHU/dpm-solver | MIT |
| Guided-Diffusion [5] | https://github.com/openai/guided-diffusion | MIT |
| Stable-Diffusion [9] | https://github.com/CompVis/stable-diffusion | CreativeML Open RAIL-M |

## D.2 Additional results for unconditional discrete-time sampling

In this section, we report on experiments with unconditional, discrete DPMs on the CIFAR10 [42] and CelebA 64x64 [43] datasets using quadratic time steps. The FID↓ scores for the PFDiff algorithm are reported for changes in the number of function evaluations (NFE) from 4 to 20. Additionally, we present FID scores on the CelebA 64x64 [43], LSUN-bedroom 256x256 [44], and LSUN-church 256x256 [44] datasets, utilizing uniform time steps. The experimental results are summarized in Table 2. Results indicate that using DDIM [20] as the baseline, our method (PFDiff) nearly achieved significant performance improvements across all datasets and NFE settings. Notably, PFDiff facilitates rapid convergence of pre-trained DPMs to the data distribution with NFE settings below 10, validating its effectiveness on discrete pre-trained DPMs and the first-order ODE solver DDIM. It is important to note that on the CIFAR10 and CelebA 64x64 datasets, we have included the FID scores of Analytic-DDIM [23], which serves as another baseline. Analytic-DDIM modifies the variance in DDIM and introduces some random noise. With NFE lower than 10, the presence of minimal random noise amplifies the error introduced by the gradient information approximation in PFDiff, reducing its error correction capability compared to the Analytic-DDIM sampler. Thus, in fewer-step sampling (NFE<10), using DDIM as the baseline is more effective than using Analytic-DDIM, which requires recalculating the optimal variance for different pre-trained DPMs, thereby introducing additional computational overhead. In other experiments with pre-trained DPMs, we validate the efficacy of the PFDiff algorithm by combining it with the overall superior performance of the DDIM solver.

Furthermore, to validate the motivation proposed in Sec. 3.2 based on Fig. 2a—that at not excessively large time step size $\Delta t$, an ODE-based solver shows considerable similarity in the noise network outputs—we compare it with the SDE-based solver DDPM [2]. Even at smaller $\Delta t$, the mean squared error (MSE) of the noise outputs from DDPM remains high, suggesting that the effectiveness of PFDiff may be limited when based on SDE solvers. Further, we adjusted the $\eta$ parameter in Eq. (6) (which controls the amount of noise introduced in DDPM) from 1.0 to 0.0 (at $\eta = 0.0$, the SDE-based DDPM degenerates into the ODE-based DDIM [20]). As shown in Fig. 2a, as $\eta$ decreases, the MSE of the noise network outputs gradually decreases at the same time step size $\Delta t$, indicating that reducing noise introduction can enhance the effectiveness of PFDiff. To verify this motivation, we utilized quadratic time steps on CIFAR10 and CelebA 64x64 datasets and controlled

Table 2: Sample quality measured by FID↓ on the CIFAR10 [42], CelebA 64x64 [43], LSUN-bedroom 256x256 [44], and LSUN-church 256x256 [44] using unconditional discrete-time DPMs, varying the number of function evaluations (NFE). Evaluated on 50k samples. PFDiff uses DDIM [20] and Analytic-DDIM [23] as baselines and introduces DDPM [2] and Analytic-DDPM [23] with $\eta = 1.0$ from Eq. (6) for comparison.

| +PFDiff | Method | NFE | | | | | | |
|---|---|---|---|---|---|---|---|---|
| | | 4 | 6 | 8 | 10 | 12 | 15 | 20 |
| CIFAR10 (discrete-time model [2], quadratic time steps) | | | | | | | | |
| × | DDPM($\eta = 1.0$) [2] | 108.05 | 71.47 | 52.87 | 41.18 | 32.98 | 25.59 | 18.34 |
| × | Analytic-DDPM [23] | 65.81 | 56.37 | 44.09 | 34.95 | 29.96 | 23.26 | 17.32 |
| × | Analytic-DDIM [23] | 106.86 | 24.02 | 14.21 | 10.09 | 8.80 | 7.25 | 6.17 |
| × | DDIM [20] | 65.70 | 29.68 | 18.45 | 13.66 | 11.01 | 8.80 | 7.04 |
| ✓ | Analytic-DDIM | 289.84 | 23.24 | 7.03 | **4.51** | **3.91** | **3.75** | **3.65** |
| ✓ | DDIM | **22.38** | **9.48** | **5.64** | 4.57 | 4.39 | 4.10 | 3.68 |
| CelebA 64x64 (discrete-time model [20], quadratic time steps) | | | | | | | | |
| × | DDPM($\eta = 1.0$) [2] | 59.38 | 43.63 | 34.12 | 28.21 | 24.40 | 20.19 | 15.85 |
| × | Analytic-DDPM [23] | 32.10 | 39.78 | 32.29 | 26.96 | 23.03 | 19.36 | 15.67 |
| × | Analytic-DDIM [23] | 69.75 | 16.60 | 11.84 | 9.37 | 7.95 | 6.92 | 5.84 |
| × | DDIM [20] | 37.76 | 20.99 | 14.10 | 10.86 | 9.01 | 7.67 | 6.50 |
| ✓ | Analytic-DDIM | 360.21 | 28.24 | 5.66 | 4.90 | 4.62 | **4.55** | **4.55** |
| ✓ | DDIM | **13.29** | **7.53** | **5.06** | **4.71** | **4.60** | 4.70 | 4.68 |
| CelebA 64x64 (discrete-time model [20], uniform time steps) | | | | | | | | |
| × | DDPM($\eta = 1.0$) [2] | 65.39 | 49.52 | 41.65 | 36.68 | 33.45 | 30.27 | 26.76 |
| × | Analytic-DDPM [23] | 102.45 | 42.43 | 34.36 | 33.85 | 30.38 | 28.90 | 25.89 |
| × | Analytic-DDIM [23] | 90.44 | 24.85 | 16.45 | 16.67 | 15.11 | 15.00 | 13.40 |
| × | DDIM [20] | **44.36** | 29.12 | 23.19 | 20.50 | 18.43 | 16.71 | 14.76 |
| ✓ | Analytic-DDIM | 308.58 | 56.04 | 14.07 | 10.98 | 8.97 | **6.39** | **5.19** |
| ✓ | DDIM | 51.87 | **12.79** | **8.82** | **8.93** | **7.70** | 6.44 | 5.66 |
| LSUN-bedroom 256x256 (discrete-time model [2], uniform time steps) | | | | | | | | |
| × | DDIM [20] | **115.63** | 47.40 | 26.73 | 19.26 | 15.23 | 11.68 | 9.26 |
| ✓ | DDIM | 140.40 | **18.72** | **11.50** | **9.28** | **8.36** | **7.76** | **7.14** |
| LSUN-church 256x256 (discrete-time model [2], uniform time steps) | | | | | | | | |
| × | DDIM [20] | 121.95 | 50.02 | 30.04 | 22.04 | 17.66 | 14.58 | 12.49 |
| ✓ | DDIM | **72.86** | **18.30** | **14.34** | **13.27** | **12.05** | **11.77** | **11.12** |

the amount of noise introduced by adjusting $\eta$, to demonstrate that PFDiff can leverage the temporal redundancy present in ODE solvers to boost its performance. The experimental results, as shown in Table 3, illustrate that with the reduction of $\eta$ from 1.0 (SDE) to 0.0 (ODE), PFDiff's sampling performance significantly improves at fewer time steps (NFE≤20). The experiment results regarding FID variations with NFE as presented in Table 3, align with the trends of MSE of noise network outputs with changes in time step size $\Delta t$ as depicted in Fig. 2a. This reaffirms the motivation we proposed in Sec. 3.2.

## D.3 Additional results for unconditional continuous-time sampling

In this section, we supplement the specific FID↓ scores for the unconditional, continuous pretrained DPMs models with first-order and higher-order ODE solvers, DPM-Solver-1, -2 and -3, [21] as baselines, as shown in Table 4. For all experiments in this section, we conducted tests on the CIFAR10 dataset [42], using the checkpoint `checkpoint_8.pth` under the

Table 3: Sample quality measured by FID↓ on the CIFAR10 [42] and CelebA 64x64 [43] using unconditional discrete-time DPMs with and without our method (PFDiff), varying the number of function evaluations (NFE) and $\eta$ from Eq. (6). Evaluated on 50k samples.

| Method | NFE | | | | | | |
|---|---|---|---|---|---|---|---|
| | 4 | 6 | 8 | 10 | 12 | 15 | 20 |
| CIFAR10 (discrete-time model [2], quadratic time steps) | | | | | | | |
| DDPM($\eta = 1.0$) [2] | 108.05 | 71.47 | 52.87 | 41.18 | 32.98 | 25.59 | 18.34 |
| +PFDiff (Ours) | 475.47 | 432.24 | 344.96 | 332.41 | 285.88 | 158.90 | 28.05 |
| DDPM($\eta = 0.5$) [20] | 71.08 | 34.32 | 22.37 | 16.63 | 13.37 | 10.75 | 8.38 |
| +PFDiff (Ours) | 432.50 | 349.09 | 311.62 | 167.65 | 59.93 | 23.17 | 10.61 |
| DDPM($\eta = 0.2$) [20] | 66.33 | 30.26 | 18.94 | 14.01 | 11.25 | 9.00 | 7.18 |
| +PFDiff (Ours) | 316.15 | 189.02 | 18.55 | 7.73 | 5.70 | 4.53 | 4.00 |
| DDIM($\eta = 0.0$) [20] | 65.70 | 29.68 | 18.45 | 13.66 | 11.01 | 8.80 | 7.04 |
| +PFDiff (Ours) | **22.38** | **9.48** | **5.64** | **4.57** | **4.39** | **4.10** | **3.68** |
| CelebA 64x64 (discrete-time model [20], quadratic time steps) | | | | | | | |
| DDPM($\eta = 1.0$) [2] | 59.38 | 43.63 | 34.12 | 28.21 | 24.40 | 20.19 | 15.85 |
| +PFDiff (Ours) | 433.25 | 439.19 | 415.41 | 317.43 | 324.58 | 326.50 | 171.41 |
| DDPM($\eta = 0.5$) [20] | 40.58 | 23.72 | 16.74 | 13.15 | 11.27 | 9.36 | 7.73 |
| +PFDiff (Ours) | 435.27 | 417.58 | 314.63 | 310.10 | 252.19 | 69.31 | 19.23 |
| DDPM($\eta = 0.2$) [20] | 38.20 | 21.35 | 14.55 | 11.22 | 9.47 | 7.99 | 6.71 |
| +PFDiff (Ours) | 394.03 | 319.02 | 45.15 | 12.71 | 7.85 | 5.10 | 4.96 |
| DDIM($\eta = 0.0$) [20] | 37.76 | 20.99 | 14.10 | 10.86 | 9.01 | 7.67 | 6.50 |
| +PFDiff (Ours) | **13.29** | **7.53** | **5.06** | **4.71** | **4.60** | **4.70** | **4.68** |

Table 4: Sample quality measured by FID↓ of different orders of DPM-Solver [21] on the CIFAR10 [42] using unconditional continuous-time DPMs with and without our method (PFDiff), varying the number of function evaluations (NFE). Evaluated on 50k samples.

| Method | order | NFE | | | | | | |
|---|---|---|---|---|---|---|---|---|
| | | 4 | 6 | 8 | 10 | 12 | 16 | 20 |
| CIFAR10 (continuous-time model [4], quadratic time steps) | | | | | | | | |
| DPM-Solver-1 [21] | 1 | **40.55** | 23.86 | 15.57 | 11.64 | 9.64 | 7.23 | 6.06 |
| +PFDiff (Ours) | 1 | 113.74 | **11.41** | **5.90** | **4.23** | **3.92** | **3.73** | **3.75** |
| DPM-Solver-2 [21] | 2 | 298.79 | 106.05 | 41.79 | 14.43 | 6.75 | **4.24** | **3.91** |
| +PFDiff (Ours) | 2 | **85.22** | **16.30** | **9.67** | **6.64** | **5.74** | 5.12 | 4.78 |
| | | | 6 | | 9 | 12 | 15 | 21 |
| DPM-Solver-3 [21] | 3 | | 382.51 | | 233.56 | 44.82 | 7.98 | **3.63** |
| +PFDiff (Ours) | 3 | | **103.22** | | **5.67** | **5.72** | **5.62** | 5.24 |

`vp/cifar10_ddpmpp_deep_continuous` configuration provided by ScoreSDE [4]. For the hyperparameter `method` of DPM-Solver [21], we adopted `singlestep_fixed`; to maintain consistency with the discrete-time model in Appendix D.2, the parameter `skip` was set to `time_quadratic` (i.e., quadratic time steps). Unless otherwise specified, we used the parameter settings recommended by DPM-Solver. The results in Table 4 show that by using the PFDiff method described in Sec. 3.4 and taking DPM-Solver as the baseline, we were able to further enhance sampling performance on

the basis of first-order and higher-order ODE solvers. Particularly, in the 6∼12 NFE range, PFDiff significantly improved the convergence issues of higher-order ODE solvers under fewer NFEs. For instance, at 9 NFE, PFDiff reduced the FID of DPM-Solver-3 from 233.56 to 5.67, improving the sampling quality by 97.57%. These results validate the effectiveness of using PFDiff with first-order or higher-order ODE solvers as the baseline.

## D.4 Additional results for classifier guidance

Table 5: Sample quality measured by FID↓ on the ImageNet 64x64 [32] and ImageNet 256x256 [32], using ADM-G [5] model with guidance scales of 1.0 and 2.0, varying the number of function evaluations (NFE). Evaluated: ImageNet 64x64 with 50k, ImageNet 256x256 with 10k samples. *We directly borrowed the results reported by AutoDiffusion [26], and AutoDiffusion requires additional search costs. *We directly borrowed the results reported by AutoDiffusion [26], and AutoDiffusion requires additional search costs. "\" represents missing data in the original paper.

| Method | Step | NFE | | | | | |
|---|---|---|---|---|---|---|---|
| | | 4 | 6 | 8 | 10 | 15 | 20 |
| ImageNet 64x64 (pixel DPMs model [5], uniform time steps, guidance scale 1.0) | | | | | | | |
| DDIM [20] | Single | 138.81 | 23.58 | 12.54 | 8.93 | 5.52 | 4.45 |
| DPM-Solver-2 [21] | Single | 327.09 | 292.66 | 264.97 | 236.80 | 166.52 | 120.29 |
| DPM-Solver-2 [21] | Multi | 48.64 | 21.08 | 12.45 | 8.86 | 5.57 | 4.46 |
| DPM-Solver-3 [21] | Single | 383.71 | 376.86 | 380.51 | 378.32 | 339.34 | 280.12 |
| DPM-Solver-3 [21] | Multi | 54.01 | 24.76 | 13.17 | 8.85 | 5.48 | 4.41 |
| DPM-Solver++(2M) [22] | Multi | 44.15 | 20.44 | 12.53 | 8.95 | 5.53 | 4.33 |
| *AutoDiffusion [26] | Single | 17.86 | 11.17 | \ | 6.24 | 4.92 | 3.93 |
| DDIM+PFDiff (Ours) | Single | **16.46** | **8.20** | **6.22** | **5.19** | **4.20** | **3.83** |
| ImageNet 256x256 (pixel DPMs model [5], uniform time steps, guidance scale 2.0) | | | | | | | |
| DDIM [20] | Single | 51.79 | 23.48 | 16.33 | 12.93 | 9.89 | 9.05 |
| DDIM+PFDiff (Ours) | Single | **37.81** | **18.15** | **12.22** | **10.33** | **8.59** | **8.08** |

In this section, we provide the specific FID scores for pre-trained DPMs in the conditional, classifier guidance paradigm on the ImageNet 64x64 [32] and ImageNet 256x256 datasets [32], as shown in Table 5. We now describe the experimental setup in detail. For the pre-trained models, we used the ADM-G [5] provided `64x64_diffusion.pt` and `64x64_classifier.pt` for the ImageNet 64x64 dataset, and `256x256_diffusion.pt` and `256x256_classifier.pt` for the ImageNet 256x256 dataset. All experiments were conducted with uniform time steps and used DDIM as the baseline [20]. We implemented the second-order and third-order methods from DPM-Solver [21] for comparison and explored the `method` hyperparameter provided by DPM-Solver for both `singlestep` (corresponding to "Single" in Table 5) and `multistep` (corresponding to "Multi" in Table 5). Additionally, we implemented the best-performing method from DPM-Solver++ [22], multi-step DPM-Solver++(2M), as a comparative measure. Furthermore, we also introduced the superior-performing AutoDiffusion [26] method as a comparison. *We directly borrowed the results reported in the original paper, emphasizing that although AutoDiffusion does not require additional training, it incurs additional search costs. "\" represents missing data in the original paper. The specific experimental results of the configurations mentioned are shown in Table 5. The results demonstrate that PFDiff, using DDIM as the baseline on the ImageNet 64x64 dataset, significantly enhances the sampling efficiency of DDIM and surpasses previous optimal training-free sampling methods. Particularly, in cases where NFE≤10, PFDiff improved the sampling quality of DDIM by 41.88%∼88.14%. Moreover, on the large ImageNet 256x256 dataset, PFDiff demonstrates a consistent performance improvement over the DDIM baseline, similar to the improvements observed on the ImageNet 64x64 dataset.

## D.5 Additional results for classifier-free guidance

In this section, we supplemented the specific FID↓ scores for the Stable-Diffusion [9] (conditional, classifier-free guidance paradigm) setting with a guidance scale ($s$) of 7.5 and 1.5. Specifically, for

the pre-trained model, we conducted experiments using the `sd-v1-4.ckpt` checkpoint provided by Stable-Diffusion. All experiments used the MS-COCO2014 [31] validation set to calculate FID↓ scores, with uniform time steps. PFDiff employs the DDIM [20] method as the baseline. Initially, under the recommended $s = 7.5$ configuration by Stable-Diffusion, we implemented DPM-Solver-2 and -3 as comparative methods, and conducted searches for the `method` hyperparameters provided by DPM-Solver as `singlestep` (corresponding to "Single" in Table 6) and `multistep` (corresponding to "Multi" in Table 6). Additionally, we introduced previous state-of-the-art training-free methods, including DPM-Solver++(2M) [22], UniPC [29], and DPM-Solver-v3(2M) [27] for comparison. The experimental results are shown in Table 6. †We borrow the results reported in DPM-Solver-v3 [27] directly. The results indicate that on Stable-Diffusion, PFDiff, using only DDIM as a baseline, surpasses the previous state-of-the-art training-free sampling methods in terms of sampling quality in fewer steps (NFE<20). Particularly, at NFE=10, PFDiff achieved a 13.06 FID, nearly converging to the data distribution, which is a 14.25% improvement over the previous state-of-the-art method DPM-Solver-v3 at 20 NFE, which had a 15.23 FID. Furthermore, to further validate the effectiveness of PFDiff on Stable-Diffusion, we conducted experiments using the $s = 1.5$ setting with the same experimental configuration as $s = 7.5$. For the comparative methods, we only experimented with the multi-step versions of DPM-Solver-2 and -3 and DPM-Solver++(2M), which had faster convergence at fewer NFE under the $s = 7.5$ setting. As for UniPC and DPM-Solver-v3(2M), since DPM-Solver-v3 did not provide corresponding experimental results at $s = 1.5$, we did not list their comparative results. The experimental results show that PFDiff, using DDIM as the baseline under the $s = 1.5$ setting, demonstrated consistent performance improvements as seen in the $s = 7.5$ setting, as shown in Table 6.

Table 6: Sample quality measured by FID↓ on the validation set of MS-COCO2014 [31] using Stable-Diffusion model [9] with guidance scales of 7.5 and 1.5, varying the number of function evaluations (NFE). Evaluated on 10k samples. †We borrow the results reported in DPM-Solver-v3 [27] directly.

| Method | Step | NFE | | | | | |
| --- | --- | --- | --- | --- | --- | --- | --- |
| | | 5 | 6 | 8 | 10 | 15 | 20 |
| MS-COCO2014 (latent DPMs model [9], uniform time steps, guidance scale 7.5) | | | | | | | |
| DDIM [20] | Single | 23.92 | 20.33 | 17.46 | 16.78 | 16.08 | 15.95 |
| DPM-Solver-2 [21] | Single | 84.15 | 74.02 | 31.87 | 17.63 | 15.15 | 13.77 |
| DPM-Solver-2 [21] | Multi | 18.97 | 17.37 | 16.29 | 15.99 | 14.32 | 14.38 |
| DPM-Solver-3 [21] | Single | 156.27 | 102.59 | 54.52 | 26.29 | 16.95 | 14.85 |
| DPM-Solver-3 [21] | Multi | 18.89 | 17.34 | 16.25 | 16.11 | 14.10 | **13.44** |
| †DPM-Solver++(2M) [22] | Multi | 18.87 | 17.44 | 16.40 | 15.93 | 15.84 | 15.72 |
| †UniPC [29] | Multi | 18.77 | 17.32 | 16.20 | 16.15 | 16.06 | 15.94 |
| †DPM-Solver-v3(2M) [27] | Multi | 18.83 | 16.41 | 15.41 | 15.32 | 15.30 | 15.23 |
| DDIM+PFDiff (Ours) | Single | **18.31** | **15.47** | **13.26** | **13.06** | **13.57** | 13.97 |
| MS-COCO2014 (latent DPMs model [9], uniform time steps, guidance scale 1.5) | | | | | | | |
| DDIM [20] | Single | 70.36 | 54.32 | 37.54 | 29.41 | 20.54 | 18.17 |
| DPM-Solver-2 [21] | Multi | 37.47 | 27.79 | 19.65 | 18.39 | 17.27 | 16.85 |
| DPM-Solver-3 [21] | Multi | 35.90 | 25.88 | 18.26 | 19.10 | 17.21 | 16.67 |
| DPM-Solver++(2M) [22] | Multi | 36.58 | 26.78 | 18.92 | 20.26 | 18.61 | 17.78 |
| DDIM+PFDiff (Ours) | Single | **24.31** | **20.99** | **18.09** | **17.00** | **16.03** | **15.57** |

## D.6 Additional ablation study results

### D.6.1 Additional results for PFDiff hyperparameters study

In this section, we extensively investigate the impact of the hyperparameters $k$ and $l$ on the performance of the PFDiff algorithm, supplemented by a series of ablation experiments regarding their configurations and outcomes. Specifically, we first conducted experiments on the CIFAR10 dataset

Table 7: Ablation of the impact of $k$ and $l$ on PFDiff in CIFAR10 [42], ImageNet 64x64 and MS-COCO2014 using DDPM [2], ScoreSDE [4], ADM-G [5] and Stable-Diffusion [9] models. We report the FID↓, varying the number of function evaluations (NFE). Evaluated: MS-COCO2014 with 10k, others with 50k samples.

| Method | NFE | | | | | |
|---|---|---|---|---|---|---|
| | 4 | 6 | 8 | 10 | 15 | 20 |
| CIFAR10 (discrete-time model [2], quadratic time steps) | | | | | | |
| DDIM [20] | 65.70 | 29.68 | 18.45 | 13.66 | 8.80 | 7.04 |
| +PFDiff-1 | 124.73 | 19.45 | 5.78 | 4.95 | 4.25 | 4.14 |
| +PFDiff-2_1 | 59.61 | **9.84** | 7.01 | 6.31 | 5.18 | 4.78 |
| +PFDiff-2_2 | 167.12 | 53.22 | 8.43 | 4.95 | **4.10** | 3.78 |
| +PFDiff-3_1 | **22.38** | 13.40 | 9.40 | 7.70 | 6.03 | 5.05 |
| +PFDiff-3_2 | 129.18 | 19.35 | **5.64** | **4.57** | 4.19 | 4.08 |
| +PFDiff-3_3 | 205.87 | 76.62 | 20.84 | 5.71 | 4.41 | **3.68** |
| CIFAR10 (continuous-time model [4], quadratic time steps) | | | | | | |
| DPM-Solver-1 [21] | **40.55** | 23.86 | 15.57 | 11.64 | 7.59 | 6.06 |
| +PFDiff-1 | 250.56 | 76.78 | 6.53 | 4.28 | **3.78** | **3.75** |
| +PFDiff-2_1 | 178.70 | **11.41** | **5.90** | 5.01 | 4.27 | 4.07 |
| +PFDiff-2_2 | 289.06 | 250.48 | 71.08 | 9.17 | 4.09 | 3.83 |
| +PFDiff-3_1 | 113.74 | 11.82 | 7.91 | 6.34 | 4.97 | 4.37 |
| +PFDiff-3_2 | 264.88 | 130.24 | 8.92 | **4.23** | **3.78** | 3.78 |
| +PFDiff-3_3 | 275.10 | 287.77 | 183.11 | 30.72 | 4.69 | 4.01 |
| ImageNet 64x64 (pixel DPMs model [5], uniform time steps, $s = 1.0$) | | | | | | |
| DDIM [20] | 138.81 | 23.58 | 12.54 | 8.93 | 5.52 | 4.45 |
| +PFDiff-1 | 26.86 | 11.39 | 7.47 | 5.83 | 4.76 | 4.39 |
| +PFDiff-2_1 | 17.14 | 8.94 | 6.38 | 5.46 | 4.30 | **3.83** |
| +PFDiff-2_2 | 23.66 | 9.93 | 6.86 | 5.72 | 4.49 | 3.94 |
| +PFDiff-3_1 | 16.74 | 9.43 | 7.19 | 5.86 | 4.69 | 4.44 |
| +PFDiff-3_2 | **16.46** | **8.20** | **6.22** | **5.19** | **4.20** | 4.28 |
| +PFDiff-3_3 | 23.06 | 9.73 | 6.92 | 5.55 | 4.47 | 4.49 |
| MS-COCO2014 (latent DPMs model [9], uniform time steps, $s = 7.5$) | | | | | | |
| DDIM [20] | 35.48 | 20.33 | 17.46 | 16.78 | 16.08 | 15.95 |
| +PFDiff-1 | 98.78 | 23.06 | **13.26** | **13.06** | 13.72 | 14.09 |
| +PFDiff-2_1 | 33.39 | **15.47** | 15.05 | 15.01 | 15.24 | 15.35 |
| +PFDiff-2_2 | 178.10 | 53.77 | 16.92 | 13.55 | **13.57** | 14.08 |
| +PFDiff-3_1 | **29.02** | 16.38 | 15.69 | 15.66 | 15.52 | 15.51 |
| +PFDiff-3_2 | 75.73 | 17.60 | 14.46 | 14.52 | 14.84 | 14.99 |
| +PFDiff-3_3 | 217.86 | 80.03 | 21.99 | 14.38 | 13.61 | **13.97** |

[42] using quadratic time steps, based on pre-trained unconditional discrete DDPM [2] and continuous ScoreSDE [4] DPMs. For the conditional DPMs, we used uniform time steps in classifier guidance ADM-G [5] pre-trained DPMs, setting the guidance scale ($s$) to 1.0 for experiments on the ImageNet 64x64 dataset [32]; for the classifier-free guidance Stable-Diffusion [9] pre-trained DPMs, we set the guidance scale ($s$) to 7.5. All experiments were conducted using the DDIM [20] algorithm as a baseline, and PFDiff-$k\_l$ configurations ($k = 1, 2, 3\ (l \leq k)$) were tested in six different algorithm configurations. The change in NFE and the corresponding FID↓ scores are shown in Table 7. The experimental results show that under various combinations of $k$ and $l$, PFDiff is able to enhance the sampling performance of the DDIM baseline in most cases across different types of pre-trained DPMs. Particularly when $k = 1$ is fixed, PFDiff-1 significantly improves the sampling performance of the DDIM baseline within the range of 8∼20 NFE. For practical applications requiring higher sampling quality at fewer NFE, optimal combinations of $k$ and $l$ can be identified by fixing NFE and sampling a small number of samples for visual analysis or computing specific metrics, without significantly

increasing the computational burden. However, as discussed in Sec. 5, although the experimental results presented in Table 7 demonstrate the excellent performance of the combinations of $k$ and $l$ under various pre-trained DPMs and NFE settings, no universally optimal configuration exists. This finding somewhat limits the generality of the proposed PFDiff algorithm and sets objectives for our future research.

### D.6.2 Ablation study of gradient guidance

To further investigate the impact of gradient guidance from the past or future on the rapid updating of current intermediate states, this section supplements experimental results and analysis using first-order and higher-order ODE solvers as baselines. Specifically, as described in Sec. 3.3, PFDiff uses a first-order ODE solver as a baseline, where future gradient guidance corrects sampling errors, with detailed proofs provided in Appendix B.2. Hence, using the first-order ODE solver DDIM [20] as a baseline, we removed past and future gradients separately and employed quadratic time steps. This was based on the pre-trained model from DDPM [2] on the CIFAR10 [42] dataset, evaluating the FID↓ metric by changing the number of function evaluations (NFE). For higher-order ODE solvers, the solving process implicitly utilizes future gradients as mentioned in Sec. 3.5, and the additional explicit introduction of future gradients increases sampling error. Therefore, when using higher-order ODE solvers as a baseline, PFDiff accelerates the sampling process using only past gradients. Specifically, for higher-order ODE solvers, we selected DPM-Solver-2 and -3 [21] as the baseline, also employing quadratic time steps, and based on the ScoreSDE [4] pre-trained model on CIFAR10 [42]. Only the future gradients were removed to validate the effectiveness of the PFDiff algorithm by changing the NFE and evaluating the FID↓ metric. As shown in Table 8, the experimental results indicate that using the first-order ODE solver DDIM as a baseline, employing only past gradients (similar to DeepCache [28]), or only future gradients, only slightly improves the baseline's sampling performance; however, combining both significantly enhances the baseline's sampling performance. Meanwhile, using higher-order ODE solvers DPM-Solver-2 and -3 as the baseline, because the algorithm inherently contains future gradients, continuing to explicitly introduce future gradients increases the overall error. Therefore, using only past gradients (PFDiff) significantly improves the baseline's sampling efficiency, especially under fewer steps (NFE<10), where PFDiff markedly ameliorates the non-convergence issues of the higher-order ODE solvers.

Table 8: Ablation of the impact of the past and future gradients on PFDiff, using different orders of ODE Solver as the baseline, in CIFAR10 [42] using DDPM [2] and ScoreSDE [4] models. We report the FID↓, varying the number of function evaluations (NFE). Evaluated on 50k samples.

| +PFDiff | Method | NFE | | | | | | |
|---|---|---|---|---|---|---|---|---|
| | | 4 | 6 | 8 | 10 | 12 | 16 | 20 |
| CIFAR10 (discrete-time model [2], quadratic time steps, baseline: 1-order ODE solver) | | | | | | | | |
| × | DDIM [20] | 65.70 | 29.68 | 18.45 | 13.66 | 11.01 | 8.80 | 7.04 |
| × | +Past (similar to [28]) | 52.81 | 27.47 | 17.87 | 13.64 | 10.79 | 8.20 | 7.02 |
| × | +Future | 66.06 | 25.39 | 11.93 | 8.06 | 6.04 | 4.17 | 4.07 |
| ✓ | +Past & Future | **22.38** | **9.84** | **5.64** | **4.57** | **4.39** | **4.10** | **3.68** |
| CIFAR10 (continuous-time model [4], quadratic time steps, baseline: 2-order ODE solver) | | | | | | | | |
| × | DPM-Solver-2 [21] | 298.79 | 106.05 | 41.79 | 14.43 | 6.75 | **4.24** | **3.91** |
| ✓ | +Past | **85.22** | **16.30** | **9.67** | **6.64** | **5.74** | 5.12 | 4.78 |
| × | +Past & Future | 351.78 | 159.13 | 57.15 | 28.24 | 15.57 | 6.47 | 4.73 |
| CIFAR10 (continuous-time model [4], quadratic time steps, baseline: 3-order ODE solver) | | | | | | | | |
| | | | 6 | | 9 | 12 | 15 | 21 |
| × | DPM-Solver-3 [21] | | 382.51 | | 233.56 | 44.82 | 7.98 | **3.63** |
| ✓ | +Past | | **103.22** | | **5.67** | **5.72** | **5.62** | 5.24 |
| × | +Past & Future | | 336.26 | | 88.99 | 27.54 | 9.59 | 5.12 |

## D.7 Inception score experimental results

To evaluate the effectiveness of the PFDiff algorithm and the widely used Fréchet Inception Distance (FID↓) metric [40] in the sampling process of Diffusion Probabilistic Models (DPMs), we have also incorporated the Inception Score (IS↑) metric [41] for both unconditional and conditional pre-trained DPMs. Specifically, for the unconditional discrete-time pre-trained DPMs DDPM [2], we maintained the experimental configurations described in Table 2 of Appendix D.2, and added IS scores for the CIFAR10 dataset [42]. For the unconditional continuous-time pre-trained DPMs ScoreSDE[4], the experimental configurations are consistent with Table 4 in Appendix D.3, and IS scores for the CIFAR10 dataset were also added. For the conditional classifier guidance paradigm of pre-trained DPMs ADM-G [5], the experimental setup aligned with Table 5 in Appendix D.4, including IS scores for the ImageNet 64x64 and ImageNet 256x256 datasets [32]. Considering that the computation of IS scores relies on features extracted using `InceptionV3` pre-trained on the ImageNet dataset, calculating IS scores for non-ImageNet datasets was not feasible, hence no IS scores were provided for the classifier-free guidance paradigm of Stable-Diffusion [9]. The experimental results are presented in Table 9. A comparison between the FID↓ metrics in Tables 2, 4, and 5 and the IS↑ metrics in Table 9 shows that both IS and FID metrics exhibit similar trends under the same experimental settings, i.e., as the number of function evaluations (NFE) changes, lower FID scores correspond to higher IS scores. Further, Figs. 1a and 1b, along with the visualization experiments in Appendix D.8, demonstrate that lower FID scores and higher IS scores correlate with higher image quality and richer

Table 9: Sample quality measured by IS↑ on the CIFAR10 [42], ImageNet 64x64 [32] and ImageNet 256x256 [32] using DDPM [2], ScoreSDE [4] and ADM-G [5] models, varying the number of function evaluations (NFE). Evaluated: ImageNet 256x256 with 10k, others with 50k samples. *We directly borrowed the results reported by AutoDiffusion [26], and AutoDiffusion requires additional search costs. "\" represents missing data in the original paper and DPM-Solver-2 [21] implementation.

| +PFDiff | Method | NFE | | | | | |
|---|---|---|---|---|---|---|---|
| | | 4 | 6 | 8 | 10 | 15 | 20 |
| CIFAR10 (discrete-time model [2], quadratic time steps) | | | | | | | |
| × | DDPM($\eta = 1.0$) [2] | 4.32 | 5.66 | 6.55 | 7.08 | 7.91 | 8.25 |
| × | Analytic-DDPM [23] | 5.76 | 6.29 | 6.93 | 7.42 | 8.07 | 8.33 |
| × | Analytic-DDIM [23] | 4.46 | 7.47 | 8.11 | 8.43 | 8.72 | 8.89 |
| × | DDIM [20] | 5.68 | 7.21 | 7.92 | 8.26 | 8.62 | 8.81 |
| ✓ | Analytic-DDIM | 1.62 | 8.78 | 9.43 | **9.61** | **9.35** | **9.29** |
| ✓ | DDIM | **7.79** | **9.29** | **9.62** | 9.43 | 9.29 | **9.29** |
| CIFAR10 (continuous-time model [4], quadratic time steps) | | | | | | | |
| × | DPM-Solver-1 [21] | **7.20** | 8.30 | 8.85 | 8.98 | 9.43 | 9.51 |
| × | DPM-Solver-2 [21] | 1.70 | 5.29 | 7.94 | 9.09 | \ | 9.74 |
| ✓ | DPM-Solver-1 | 4.29 | **9.25** | **9.76** | **9.86** | **9.85** | **9.97** |
| ✓ | DPM-Solver-2 | 6.96 | 8.58 | 8.75 | 9.26 | \ | 9.69 |
| ImageNet 64x64 (pixel DPMs model [5], uniform time steps, guidance scale 1.0) | | | | | | | |
| × | DDIM [20] | 7.02 | 31.13 | 40.51 | 46.06 | 54.37 | 59.09 |
| × | DPM-Solver-2(Multi) [21] | 19.03 | 33.75 | 44.65 | 51.79 | 62.18 | 67.69 |
| × | DPM-Solver-3(Multi) [21] | 17.46 | 29.80 | 41.86 | 50.90 | 62.68 | 68.44 |
| × | DPM-Solver++(2M) [22] | 20.72 | 34.22 | 43.62 | 50.02 | 60.00 | 65.66 |
| × | *AutoDiffusion [26] | 34.88 | 43.37 | \ | 57.85 | 64.03 | 68.05 |
| ✓ | DDIM | **35.67** | **50.14** | **58.42** | **59.78** | **64.54** | **69.09** |
| ImageNet 256x256 (pixel DPMs model [5], uniform time steps, guidance scale 2.0) | | | | | | | |
| × | DDIM [20] | 37.72 | 95.90 | 122.13 | 144.13 | 165.91 | 179.27 |
| ✓ | DDIM | **55.90** | **122.56** | **158.57** | **169.72** | **183.07** | **192.70** |

details generated by the PFDiff sampling algorithm. These results further confirm the effectiveness of the PFDiff algorithm and the FID metric in evaluating the performance of sampling algorithms.

## D.8 Additional visualize study results

To demonstrate the effectiveness of PFDiff, we present the visual sampling results on the CIFAR10 [42], CelebA 64x64 [43], LSUN-bedroom 256x256 [44], LSUN-church 256x256 [44], ImageNet 64x64 [32], ImageNet 256x256 [32], and MS-COCO2014 [31] datasets in Figs. 5-10. These results illustrate that PFDiff, using different orders of ODE solvers as a baseline, is capable of generating samples of higher quality and richer detail on both unconditional and conditional pre-trained Diffusion Probabilistic Models (DPMs).

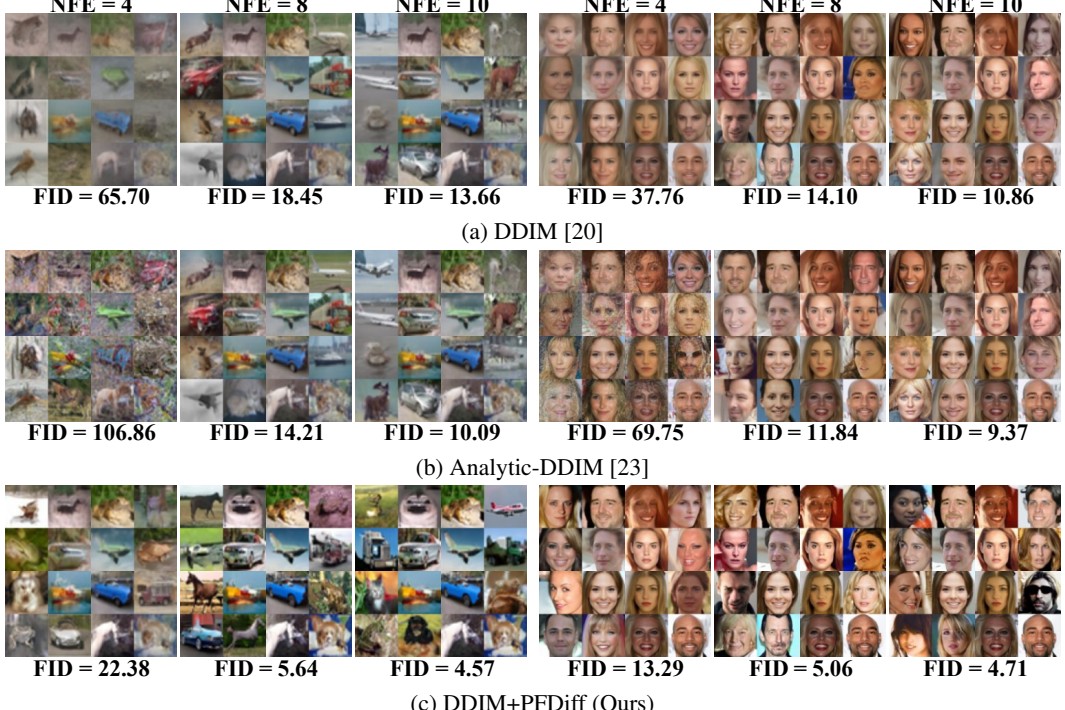

Figure 5: Random samples by DDIM [20], Analytic-DDIM [23], and PFDiff (baseline: DDIM) with 4, 8, and 10 number of function evaluations (NFE), using the same random seed, quadratic time steps, and pre-trained discrete-time DPMs [2, 20] on CIFAR10 [42] (left) and CelebA 64x64 [43] (right).

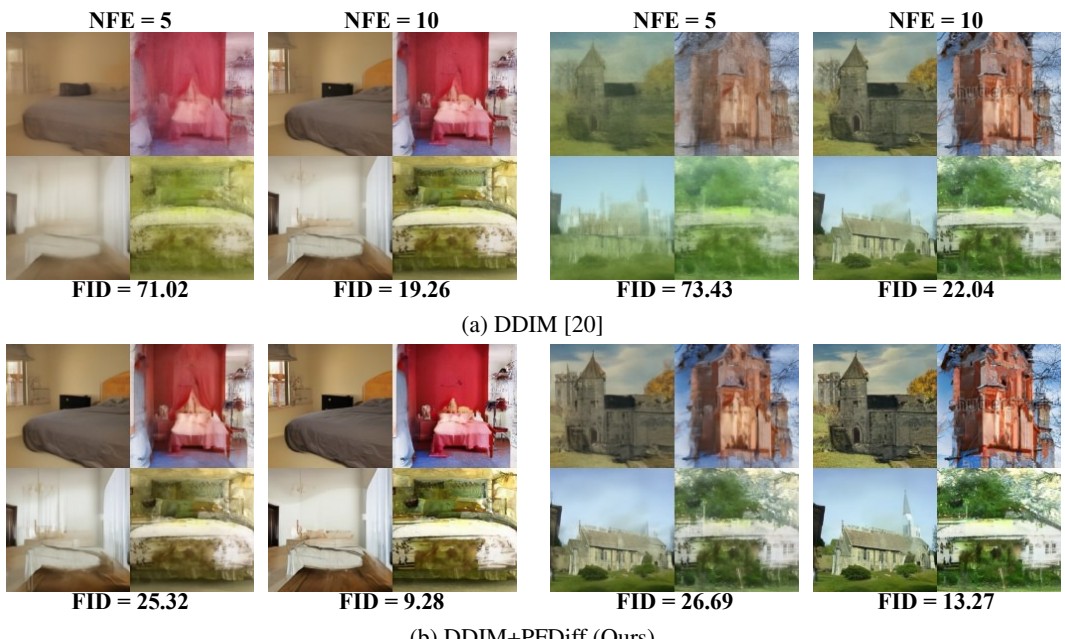

Figure 6: Random samples by DDIM [20] and PFDiff (baseline: DDIM) with 5 and 10 number of function evaluations (NFE), using the same random seed, uniform time steps, and pre-trained discrete-time DPMs [2] on LSUN-bedroom 256x256 [44] (left) and LSUN-church 256x256 [44] (right).

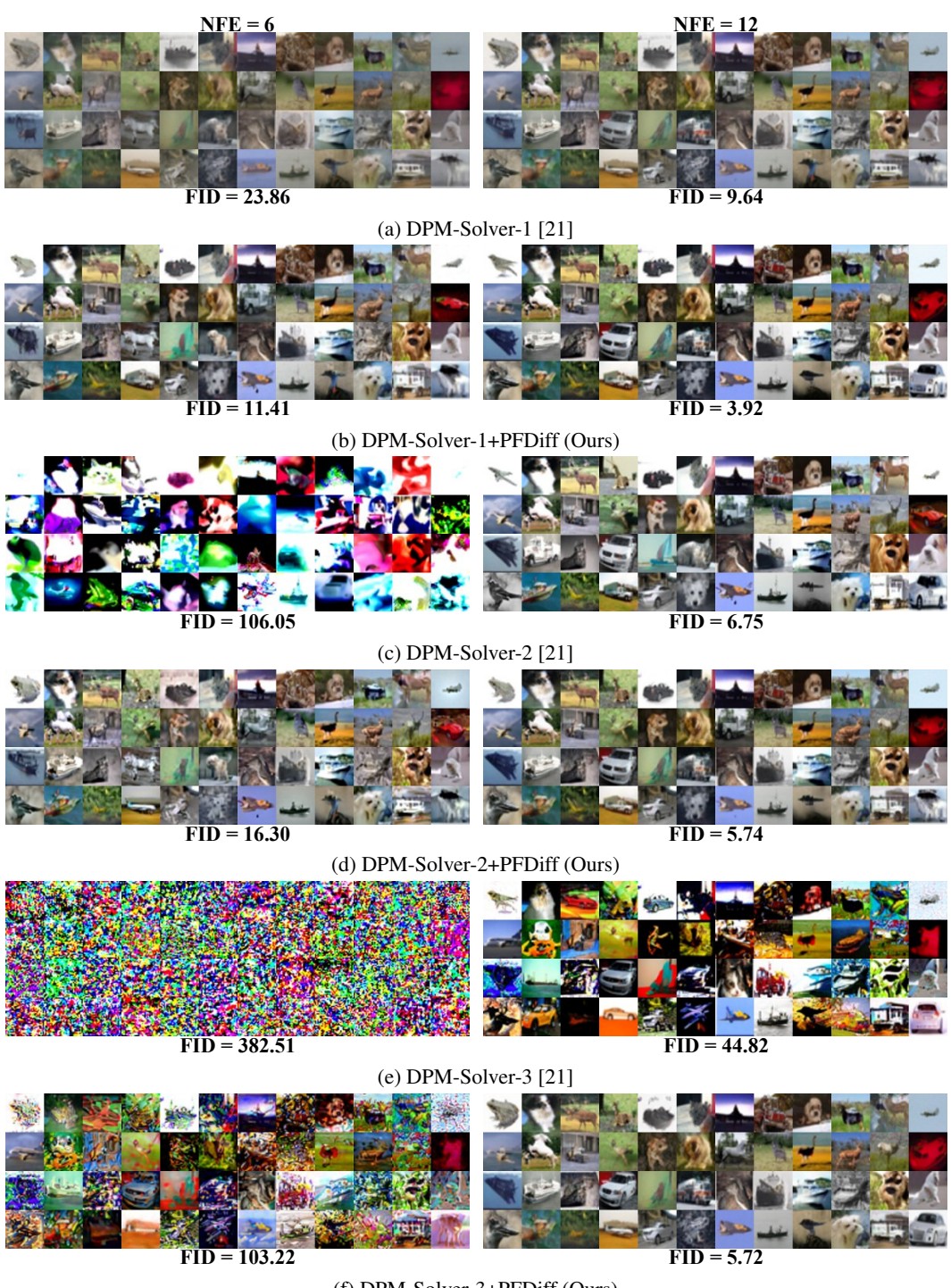

Figure 7: Random samples by DPM-Solver-1, -2, and -3 [21] with and without our method (PFDiff) with 6 and 12 number of function evaluations (NFE), using the same random seed, quadratic time steps, and pre-trained continuous-time DPMs [4] on CIFAR10 [42].

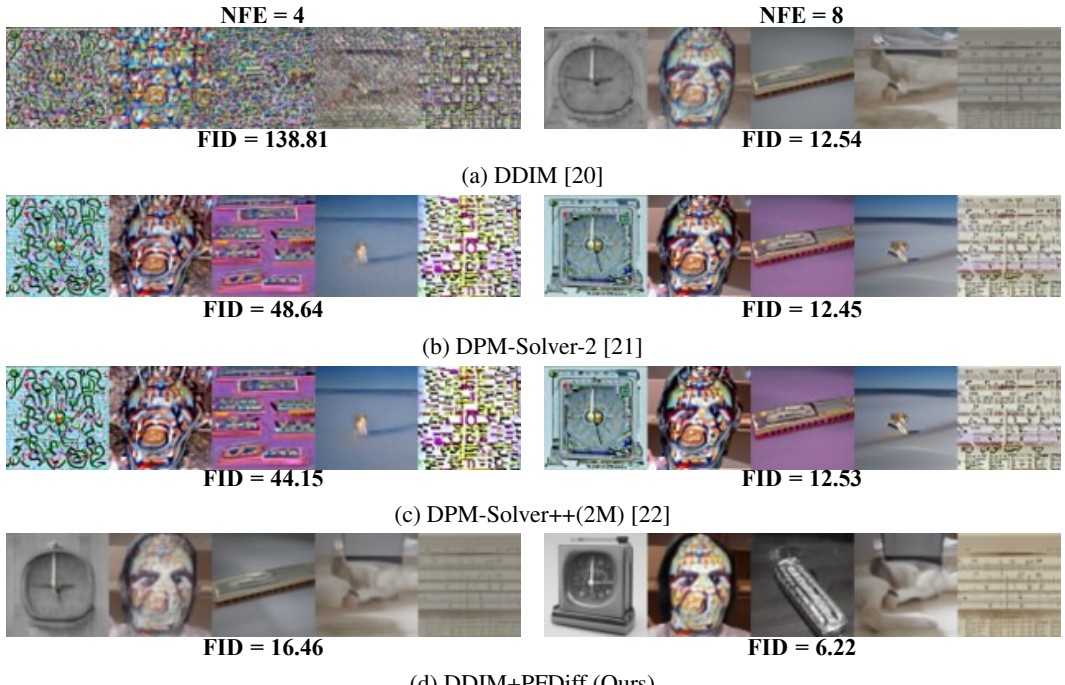

Figure 8: Random samples by DDIM [20], DPM-Solver-2 [21], DPM-Solver++(2M) [22], and PFDiff (baseline: DDIM) with 4 and 8 number of function evaluations (NFE), using the same random seed, uniform time steps, and pre-trained Guided-Diffusion [5] on ImageNet 64x64 [32] with a guidance scale of 1.0.

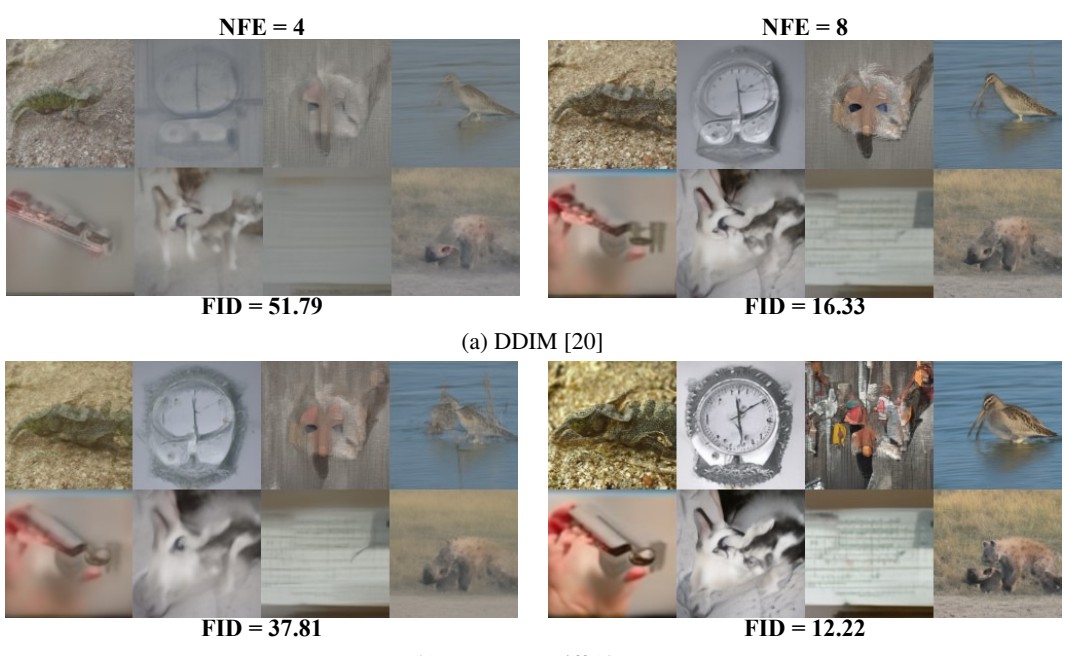

Figure 9: Random samples by DDIM [20] and PFDiff (baseline: DDIM) with 4 and 8 number of function evaluations (NFE), using the same random seed, uniform time steps, and pre-trained Guided-Diffusion [5] on ImageNet 256x256 [32] with a guidance scale of 2.0.

**Text Prompts** (listed from left to right):
A large bird is standing in the water by some rocks.
A candy covered cup cake sitting on top of a white plate.
People at a wine tasting with a table of wine bottles and glasses of red wine.
A bathtub sits on a tiled floor near a sink that has ornate mirrors over it while greenery grows on the other side of the tub.
A kitchen and dining area in a house with an open floor plan that looks out over the landscape from a large set of windows.

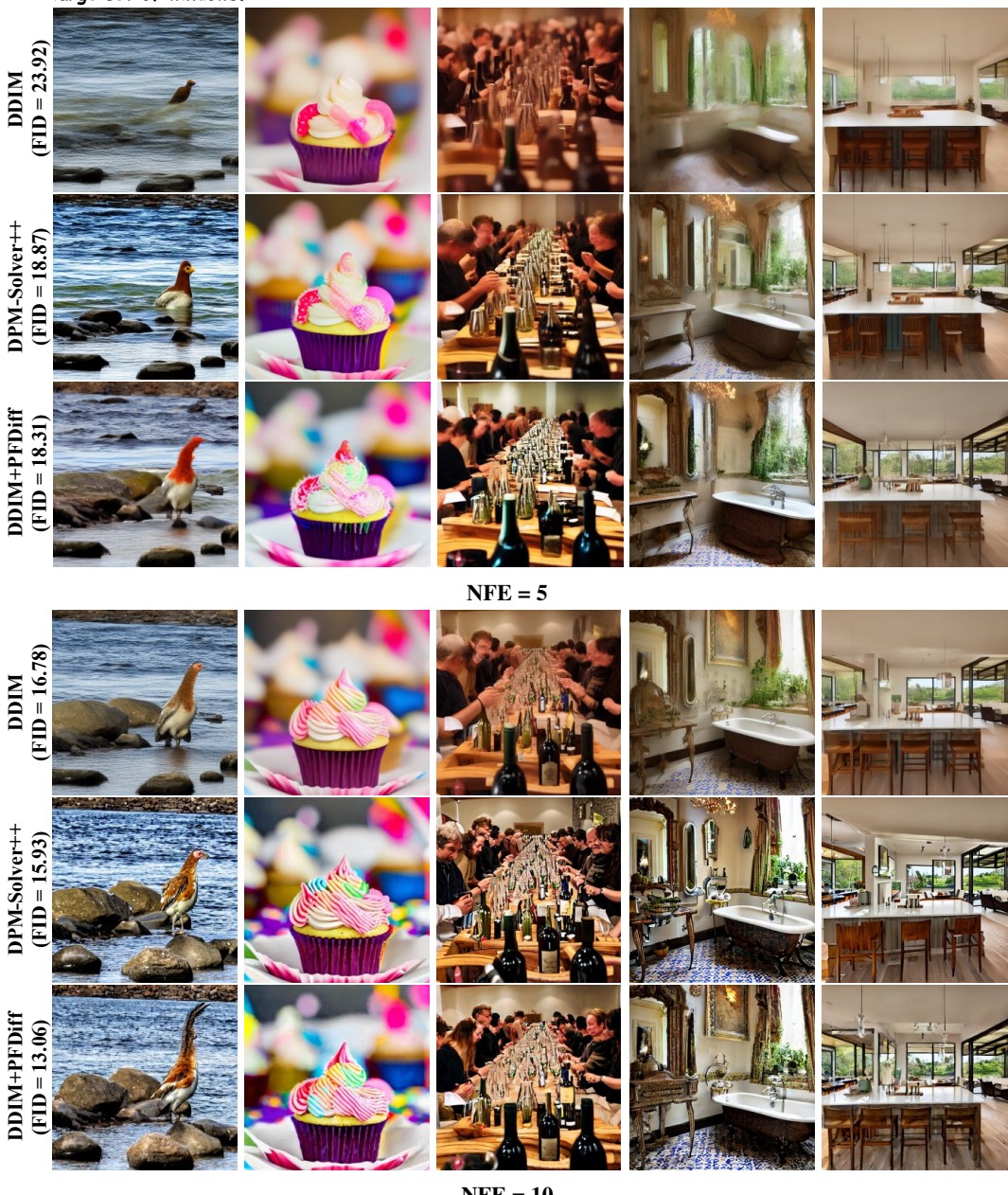

Figure 10: Random samples by DDIM [20], DPM-Solver++(2M) [22], and PFDiff (baseline: DDIM) with 5 and 10 number of function evaluations (NFE), using the same random seed, uniform time steps, and pre-trained Stable-Diffusion [9] with a guidance scale of 7.5. Text prompts are a random sample from the MS-COCO2014 [31] validation set.

