# OpenReview forum: "PFDiff: Training-free Acceleration of Diffusion Models through the Gradient Guidance of Past and Future"
_NeurIPS.cc/2024/Conference — Submitted to NeurIPS 2024_

### Official Review · Reviewer_FCeN · 2024-07-08

**Soundness:** 3
**Presentation:** 4
**Contribution:** 2
**Rating:** 4
**Confidence:** 3

**Summary:**

To accelerate the sampling speed in diffusion models, this paper proposes a training-free denoising method, dubbed PFDiff.
Concretely, PFDiff employs the gradient from past time steps to update intermediate states, aiming to reduce unnecessary NFEs while correcting for discretization errors.
In this manner, PFDiff enables to improve classic samplers without any training computation.
Importantly, experimental results demonstrate the effectiveness of the proposed PFDiff.

**Strengths:**

1. Reducing discretization errors in diffusion models in a training-free manner is attractive and practical.

2. The motivation of using previous gradients to guide the current sampling direction is intuitively plausible, and the proposed method is technically sound.

3. The presentation is excellent and the figures all are readable.

**Weaknesses:**

1. In my humble opinion, the theoretical analysis part is naive. Can you provide more explanation about why previous gradients is helpful to guide current sampling direction? Since different noise levels correspond to different gradients, is there any harm in denoising images with the proposed method?

2. Many works investigate using previous gradients to improve sampling speed, so the contribution is limited.

**Questions:**

1. Is there any memory overhead for saving the gradients?

2. Can you present more experiments on ImageNet 256? Including conditional and unconditional.

3. How about the comparison with DEIS sampler?

4. Can the proposed method connect with distillation models?

5. If more metrics are tested, such as recall and precision, it would be better.

**Limitations:**

Please see in Weaknesses and Questions. If all of my concerns are addressed, I will improve my score.

---

> ### Author Rebuttal · Authors · 2024-08-06
>
> We sincerely appreciate the reviewer's valuable suggestions.
>
> ***W1: Previous gradient guided sampling and PFDiff's harm analysis.***
>
> **A**: Let's start with a possible misunderstanding. In PFDiff, previous gradients do not assist in guiding the sampling direction; rather, it is the future gradients that have a more significant guiding effect on the current state. Our primary motivation for replacing current gradients with past ones is their high similarity (see Fig. 2(a)), which reduces computational costs by skipping the current gradient computing. Regarding future gradients, we found that the optimal gradient guiding the current state exists at a future moment, not the current (Appendix B.2, lines 508-521, and **response to reviewer sKuL _W2_**). So using future gradients to approximate the optimal gradient offers better guidance for sampling the current state.
>
> **PFDiff does not harm image denoising.** Since the denoising process involves solving discretized SDEs/ODEs, guiding current state sampling with the current gradient introduces unavoidable discretization errors, especially with fewer NFEs. PFDiff addresses this by using future gradients to approximate the optimal gradient, reducing discretization errors and accelerating sampling without harming quality.
>
> ***W2: Many works investigate using previous gradients to improve sampling speed.***
>
> **A**: Using previous gradients alone (either entirely replacing or partially caching) to guide the current state's sampling is inefficient (see **common response, Q3, Table C**). The efficiency of the PFDiff comes from its information-efficient sampling update process, which involves the current state, past, and future gradients. **PFDiff completes two updates with just one gradient computation (1 NFE), which is equivalent to achieving an update process of a second-order ODE solver with 2 NFE**. so omitting either future or past gradients would significantly limit efficiency. PFDiff shows significant differences in improving sampling speed compared to many existing methods that utilize previous gradients.
>
> ***Q1: Is there any memory overhead for saving the gradients?***
>
> **A**: **The gradient saving of our PFDiff does not lead to memory overhead.** This is because each update overwrites the previously saved gradients ( line 8 of Algorithm 1). Only one gradient needs to be stored at any given time, which is equivalent to the memory required to store one image.
>
> ***Q2: More experiments on ImageNet 256, conditional and unconditional.***
>
> **A**: Yes, we added experiments on the ImageNet 256 as follows:
>
> Unconditional, ImageNet 256, FID↓
>
> | Method\NFE  | 4         | 6         | 8         | 10        | 15        | 20        |
> | ----------- | --------- | --------- | --------- | --------- | --------- | --------- |
> | DDIM        | 75.27     | 46.01     | 34.67     | 28.52     | 23.09     | 20.89     |
> | DDIM+PFDiff | 64.57| 37.10 | 26.01 | 21.24 | 18.44| 17.45 |
>
> Conditional (s=2.0), ImageNet 256, FID↓
>
> | Method\NFE  | 4         | 6         | 8         | 10        | 15       | 20       |
> | ----------- | --------- | --------- | --------- | --------- | -------- | -------- |
> | DDIM        | 51.79     | 23.48     | 16.33     | 12.93     | 9.89     | 9.05     |
> | DDIM+PFDiff | 37.81 | 18.15| 12.22 | 10.33| 8.59 | 8.08 |
>
> As shown above, whether in conditional or unconditional experiments on the ImageNet 256 dataset, PFDiff consistently enhances the performance of DDIM. This further validates the effectiveness and wide applicability of PFDiff.
>
>
> ***Q3: How about the comparison with DEIS sampler?***
>
> **A**: We added a comparison with the DEIS sampler on the CelebA 64x64 dataset, utilizing the default $t$AB3 version from the DEIS [1] codebase, and kept other experimental settings consistent with our PFDiff. The specific experimental results are as follows:
>
> Unconditional, CelebA 64x64, FID↓
>
> | Method\NFE      | 4         | 6        | 8        | 10       | 12       | 15       | 20       |
> | --------------- | --------- | -------- | -------- | -------- | -------- | -------- | -------- |
> | DDIM            | 37.76     | 20.99    | 14.10    | 10.86    | 9.01     | 7.67     | 6.50     |
> | $t$AB3-DEIS [1] | 27.33     | 14.76    | 9.30     | 6.38     | 4.96     | **4.18** | **3.32** |
> | DDIM+PFDiff     | **13.29** | **7.53** | **5.06** | **4.71** | **4.60** | 4.70     | 4.68     |
>
> Like in the table, under conditions of fewer NFE, PFDiff outperforms DEIS. Particularly, at NFE=10, the PFDiff shows a faster convergence rate than DEIS. This further validates the efficiency and superiority of PFDiff under low NFE conditions. In the revised manuscript, we have incorporated the experimental results into Table 2 of the Appendix and Fig. 4(c).
>
>
> ***Q4: Can the proposed method connect with distillation models?***
>
> **A**: It is non-trivial as PFDiff involves future gradients. However, theoretically, we can distill model information onto the temporal scales of PFDiff to achieve model distillation. E.g., using model distillation instead of learning optimal future gradients guided by hyperparameters $k$ and $l$ for updating PFDiff. This will be a focus of our future research.
>
> ***Q5: More experimental metrics.***
>
> **A**: We added more experiments and compared them using recall, precision, and sFID metrics (ImageNet results added in Appendix D.7).
>
> Unconditional, CIFAR10
>
> | Method\NFE  | 10    |       |         |            | 20   |       |         |            |
> | ----------- | ----- | ----- | ------- | ---------- | ---- | ----- | ------- | ---------- |
> |             | FID↓  | sFID↓ | Recall↑ | Precision↑ | FID↓ | sFID↓ | Recall↑ | Precision↑ |
> | DDIM        | 13.66 | 8.10  | 50.87   | 62.29      | 7.04 | 5.43  | 56.06   | 64.95      |
> | DDIM+PFDiff | 4.57  | 4.43  | 59.85   | 65.54      | 3.68 | 4.20  | 59.84   | 66.23      |
>
> As shown in the table, PFDiff consistently improved performance across all metrics, which shows the effectiveness of the PFDiff.

---

### Official Review · Reviewer_sKuL · 2024-07-12

**Soundness:** 3
**Presentation:** 2
**Contribution:** 2
**Rating:** 5
**Confidence:** 5

**Summary:**

This paper proposes PFDiff, a fast training-free sampler for diffusion models. PFDiff updates the current state with both the past score network evaluation and the future score network evaluation. It can achieve good sample quality with less than 10 NFE. The authors showcase the effectiveness of PFDiff on various pre-trained diffusion models.

**Strengths:**

1. With proper tuning, the proposed PFDiff can outperform existing ODE solvers in the low-NFE regime on various datasets.
2. The authors provide comprehensive technical details about the proposed algorithm.

**Weaknesses:**

1. Flawed justification for future gradient: The authors' claim that using future gradient information is better than using current gradient information is based on the mean value theorem (lines 164-167 and Appendix B.2). However, this theorem only guarantees the existence of an optimal point within an interval, not its specific location. Therefore, the mean value theorem itself doesn't justify the preference for future gradients.
2. Missing justification for the approximation: While the authors claim that their approximation is better, there is no theoretical justification for it. The proof in Appendix B.2 assumes that the optimal point is already known, which is not informative. The manuscript will benefit from a further approximation error analysis.
3. Expensive and case-specific tuning: the proposed method essentially defines a set of candidate points and searches for the optimal point by tuning parameters $k$ and $l$. This tuning process can be computationally expensive and needs to be done for each specific case, limiting its practicality.

**Questions:**

Line 163-164: the mean value theorem always holds for any continuous differentiable function. What does it even mean for the mean value theorem to hold "approximately"? Also, it's unclear how the observations mentioned in lines 161-163 indicate the mean value theorem.

**Limitations:**

The algorithm performance depends heavily on parameters $k$ and $l$ as shown in Table 7.  Optimal values for $k$ and $l$ vary based on the pre-trained model and the number of function evaluations. This necessitates extensive parameter tuning when applying the proposed method in practice.

---

> ### Author Rebuttal · Authors · 2024-08-06
>
> Thanks for the valuable comments, they help improve our paper.
>
> ***W1: Flawed justification for future gradient.***
>
> **A:** Thanks for pointing out that the simplified description (lines 164-167) may have led to some misunderstanding. We now provided a more comprehensive explanation of the underlying logic (**Approximation error analysis, please see _W2_ below**).
>
> - Firstly, the mean value theorem ensures the existence of an optimal point within an interval.
> - Secondly, as analyzed in lines 508-515 and 519-521, the sampling trajectory of DPMs is not a simple linear relation (if it were a straight line, a larger sampling step size would not decrease the sampling quality), thus deducing that the optimal point would not be at the interval's endpoints. Therefore, sampling using the gradient at the current time point is not optimal.
> - More importantly, we introduce hyperparameters $k$ and $l$ to approximate the optimal point through searching, as shown in Table 7. Even without precisely pinpointing the optimal point, adjusting these parameters has significantly improved the performance of PFDiff over the baseline. Therefore, our conclusion does not solely rely on theoretical derivation; both the mean value theorem and experimental results collectively support the viewpoint that using the gradient corresponding to future time points results in smaller discretization errors than using current gradients. We have made appropriate modifications to lines 164-167 and Appendix B.2 based on the above discussion.
>
> ***W2: Approximation error analysis.***
>
> **A: Great suggestion !** We have added the following error analysis into Appendix B.2:
>
> Starting from Eq. (8):
>
> $x_{t_i}=x_{t_{i-1}}+\int_{t_{i-1}}^{t_i} s(\epsilon_\theta(x_t,t),x_t,t) \mathrm{d}t.$
>
> We define $s_{\theta }(x_{t}, t):=s(\epsilon_\theta(x_t,t),x_t,t)$, and further analyze the term that may cause errors, $\int_{t_{i-1}}^{t_i} s_{\theta }(x_{t}, t) \mathrm{d}t$. Applying Taylor's expansion at $t=r, r\in [t_{i-1},t_{i}]$, we derive:
>
> $\int_{t_{i-1}}^{t_i} s_\theta\left(x_t, t\right) d t=\int_{t_{i-1}}^{t_i}\left[\sum_{n=0}^{\infty} \frac{s_\theta^{(n)}\left(x_{r}, r \right)}{n!}(t-r)^n+R_n(t)\right] dt \approx
> \frac{1}{(n+1)!} \sum_{n=0}^{\infty} s_\theta^{(n)}\left(x_{r}, r\right)\left[\left(t_i-r\right)^{n+1}-\left(t_{i-1}-r\right)^{n+1}\right].$
>
> Furthermore, we analyze $r=t_{i-1}$ (i.e., the gradient corresponding to the current time point), and $r\in (t_{i-1},t_{i})$ (i.e., the gradient corresponding to the future time point). We compare the absolute values of the coefficients of the higher-order derivative terms corresponding to $r=t_{i-1}$ and $r\in (t_{i-1},t_{i})$, namely $|(t_i-t_{i-1})^n|$ and $| (t_{i}-r)^n -(t_{i-1}-r)^n |$, where $r \in (t_{i-1},t_{i})$,  $n \ge 2$ and $ | t_{i}-r  |+ | t_{i-1}-r  |= | t_i - t_{i-1} |$.
>
> 1. When $n$ is even, we can infer $\left | (t_{i}-r)^n -(t_{i-1}-r)^n \right |= |\ | t_{i}-r |^n - |t_{i-1}-r |^n | $. Furthermore, due to $ | t_{i}-r  | < | t_i - t_{i-1}|$ and $ | t_{i-1}-r  | <  | t_i - t_{i-1} |$, we can infer that $ |\  | t_{i}-r |^n - |t_{i-1}-r |^n  | <\max(| t_i-r |^n, |t_{i-1}-r|^n)<  | (t_i-t_{i-1})^n |$ holds.
>
> 2. When $n$ is odd, we can infer $ |(t_{i}-r)^n -(t_{i-1}-r)^n|=| t_{i}-r | ^n + | t_{i-1}-r | ^n$, where $n\ge 3$. Let $a= | t_{i}-r  |$, $b= | t_{i-1}-r  |$ and $c=| t_i - t_{i-1}|$, we have $a,b,c>0$ and $c>a,b$. Next, using mathematical induction, we prove $a^n + b^n <c^n$, where $n\ge 3$ and $a+b=c$.
>    * When $n=3$，$c^3 = (a+b)^3=a^3 + 3 a^2 b + 3 a b^2 + b^3 > a^3 + b^3$, hold.
>    * When $n=k$ ($k\ge 3$, $k \in \mathbb{N}$), suppose $a\le b$, then $a^k + b^k < c^k$ hold. When $n=k+1$:
>      $a^{k+1} +b^{k+1}=a\cdot a^k+b \cdot b^k\le b\cdot a^k+b \cdot b^k=b \cdot (a^k + b^k)<b \cdot c^k<c^{k+1}$. Overall, $a^n + b^n <c^n$ holds, thus $ | t_{i}-r   | ^n + | t_{i-1}-r  | ^n < | t_i - t_{i-1} |^n$ holds.
>
> In summary, we find $| (t_{i}-r)^n -(t_{i-1}-r)^n | < |(t_i-t_{i-1})^n|$, where $r\in (t_{i-1},t_{i})$. Firstly, this demonstrates that using the future time point $r$ gradient compared to the current time point $t_{i-1}$ gradient, the absolute values of the coefficients for higher-order derivative terms in the Taylor expansion are smaller. Secondly, as is well known, discretizing Eq. (8) neglects higher-order derivative terms, thereby introducing discretization errors. Finally, these suggest that neglecting higher-order terms has less impact when sampling with future gradients, further demonstrating that PFDiff's use of future gradient approximations in place of optimal gradients results in smaller sampling errors.
>
> ***W3 and L1: Tuning hyperparameters k and l is both expensive and highly case-specific.***
>
> **A**: As in *common response, Q1*, searching for $k$ and $l$ does not necessitate extensive parameter tuning. Moreover, even without conducting a search and simply fixing $k=1$ and $l=1$, PFDiff still significantly enhances the baseline's sampling performance.
>
> ***Q1: Two questions regarding the Mean Value Theorem.***
>
> **A**: Firstly, regarding lines 161-163, we cite from [1], which concludes that the sampling trajectories of DPMs' ODE solvers "almost" lie in a two-dimensional plane embedded in a high-dimensional space. This ensures the applicability of the mean value theorem in the context of ODE solutions for DPMs.
>
> Secondly, we revised lines 163-164 and replaced "hold 'approximately' " with more explanation to avoid any misunderstandings. **We intend to convey that**: It is well known that the mean value theorem for real-valued functions does not hold in the case of vector-valued functions. However, given the previously mentioned conclusions, the unique geometric property where the sampling trajectories of DPMs' ODE solvers "almost" lie in a two-dimensional subspace ensures the applicability of the mean value theorem in the ODE solving process of DPMs.
>
> [1] Zhenyu Zhou et al., Fast ode-based sampling for diffusion models in around 5 steps, CVPR 2024.

---

> > ### Comment · Reviewer_sKuL · 2024-08-13
> >
> > Thanks for the detailed rebuttal. It addressed most of my concerns. I'm leaning toward borderline acceptance now. I appreciate the effort in approximation analysis. However, I think there is an issue in the analysis.
> > - The new approximation error analysis is based on the absolute values of the coefficients of higher-order derivatives. It does not necessarily imply that the approximation error $\int R_n(t)dt$ is smaller. In fact, using the current gradient could be optimal. For example, consider the case where $s_{\theta}(x_t, t)=1+4t-6t^2$ and $t_i=1, t_{i-1}=0$. The approximation error based on the current point $\int_0^1s_{\theta}(x_t, t)dt=s_{\theta}(0)\int_0^1dt=1$ is exact.

---

> > > ### Author Response · Authors · 2024-08-13
> > >
> > > We sincerely appreciate your consideration of raising the rating. In terms of the new issue in the analysis, first, we want to emphasize that utilizing current gradients introduces two discretization errors: **neglecting remainder terms and neglecting higher-order derivative terms**. The future gradients aim to reduce the impact of neglecting higher-order derivative terms, rather than remainder term $\int_{t_{i-1}}^{t_i} R_n(t)dt$.
> > >
> > > We provide a more detailed response here:
> > >
> > > We carefully checked the example you provided, and found it is just a "**coincidence**" that occurs in low-dimensional, low-order functions. In our proof process, there is a step involving approximation:
> > >
> > > $\int_{t_{i-1}}^{t_i}\left[\sum_{n=0}^{\infty} \frac{s_\theta^{(n)}\left(x_{r}, r \right)}{n!}(t-r)^n+R_n(t)\right] dt \approx
> > > \frac{1}{(n+1)!} \sum_{n=0}^{\infty} s_\theta^{(n)}\left(x_{r}, r\right)\left[\left(t_i-r\right)^{n+1}-\left(t_{i-1}-r\right)^{n+1}\right].$
> > >
> > > Whether using current or future gradients, the impact of the remainder term $\int_{t_{i-1}}^{t_i} R_n(t)dt$ is neglected. Further, in the discretization of integrals for a first-order ODE solver, higher-order derivative terms (containing $s_\theta^{(n)}\left(x_{r}, r \right), n \ge 1$) are directly neglected, thus our focus is on analyzing the impact of higher-order derivative terms. In our previous response, we demonstrated using future gradients can reduce the discretization errors caused by neglecting higher-order terms (since the coefficients' absolute values are smaller).
> > >
> > > Regarding the further analysis of the example $s_{\theta }(x_{t}, t)=1+4t-6t^{2}$. **Firstly**, based on the future time point $t=\frac{2}{3}$ (i.e., PFDiff-2_2), $\int_{0}^{1} s_{\theta }(x_{t}, t) dt=s_{\theta }(\frac{2}{3} )\int_{0}^{1} dt=1$ is also exact. **Secondly**, when $n\ge 3$, the higher-order derivative terms in the example are all zero. However, in the sampling process of DPMs, this corresponds to the derivatives of high-dimensional neural networks $s_\theta^{(n)}\left(x_{r}, r\right)$, which generally are not zero. Therefore, the future gradients are dedicated to complex neural network functions in practical applications rather than low-dimensional, low-order functions. **Lastly**, the "correct" solution is merely a "coincidence" in the example, as the current gradients neglect rather than effectively address the errors from remainder terms and higher-order derivative terms. The interaction of two errors leads to the "correct" result.
> > >
> > > We have updated the proof process in the paper based on the above points. We are very pleased to discuss If you have further questions. Thank you again for your response!

---

> > > > ### Comment · Reviewer_sKuL · 2024-08-13
> > > >
> > > > Thanks for the response. I've raised my score to 5.
> > > >
> > > > Regarding your latest response, the approximation based on future time might be more accurate for neural networks in practice. But I think the proof is not rigorous. One can always construct a higher-dimensional and higher-order function example like the one constructed above. We can add as many higher-order terms as we want, such as the current gradient is optimal. For example, $s_{\theta}(t)=1+ \sum_{k=1}^{K} (2kx^{2k-1}-(2k+1)x^{2k})$. The approximation based on the current time step is always exact, but PFDiff with finite order is not always exact depending on the choice of $r$.

---

> > > > > ### Author Response · Authors · 2024-08-14
> > > > >
> > > > > Thank you again for your response and for raising the rating!
> > > > >
> > > > > Our proof shows that future gradients can reduce discretization errors caused by neglecting higher-order terms, but the errors from remainder terms still exist. We would like to emphasize that the current gradients **neglect, rather than effectively address,** the errors from remainder terms and higher-order derivative terms. In a few cases, The interaction of the two discretization errors cancels each other out, leading to the exact result. In practical applications of DPMs sampling, such situations generally do not occur due to the complexity of neural network functions.
> > > > >
> > > > > Thank you again for the counterexamples you provided. We will further elaborate such special cases in the proof process of our manuscript. We are very pleased to discuss if you have any further questions!

---

### Official Review · Reviewer_NAJn · 2024-07-12

**Soundness:** 3
**Presentation:** 2
**Contribution:** 4
**Rating:** 6
**Confidence:** 4

**Summary:**

The paper proposes PFDiff, a training-free approach for accelerating diffusion models. Motivated by the high similarity of the diffusion network outputs at adjacent timesteps on the sampling trajectory, PFDiff utilizes past and future information for sampling with time-skipping,
and decreases the number of function evaluations (NFEs) significantly. Experiments on various settings show significant acceleration, especially in the low NFE regime.

**Strengths:**

- The method is training-free and can be plug into existing solvers.
- The motivation and overall method seem reasonable.
- The improvements is significant especially in the low NFE regime. State-of-the-art diffusion solvers like UniPC and DPM-Solver-v3 are compared.
- The finding that first-order solver (DDIM), along with PFDiff, can outperform high-order solvers, is intriguing.

**Weaknesses:**

- The highly concise writing and complex notations might be a bit confusing. Additional illustrations for certain local algorithm procedures can be helpful for understanding the overall idea.
- There are fundamental mistakes in the writing. Eqn. (8) (9) are represented as Euler discretizations of the original PF-ODE. However, both DDIM and the series of DPM-Solvers rely on exponential integrators to transform the PF-ODE into other forms, so that the linear term $x_t$ is cancelled. Though this does not mean the method is wrong, such simplified writing can be misleading. The authors are obligated to correct this, or I will be forced to reject this paper.
- It will be more convincing to include experiments on EDM, the SOTA diffusion model on CIFAR-10 and ImageNet 64x64.

**Questions:**

- Are there any insights why PFDiff is more effective on first-order solvers (DDIM+PFDiff even outperforms high-order solvers)?

**Limitations:**

Yes.

---

> ### Author Rebuttal · Authors · 2024-08-06
>
> We sincerely appreciate the reviewer's valuable review of the manuscript and the recognition of our work of the work presented in the paper. Below are our responses to all questions. We kindly hope you could consider increasing the score if you are satisfied.
>
> ***W1: The highly concise writing and complex notations might be a bit confusing.***
>
> **A**: Thanks to the reviewer for pointing this out this issue. As now clarified in the *common response, Q2,* and the one-page PDF attachment, we gave more explanation of the notations and, crucially, added flowcharts for the core iterative processes of the PFDiff algorithm. We hope this will help readers better understand our algorithm.
>
> ***W2: There are fundamental mistakes in the writing.***
>
> **A**: We appreciate the reviewer for pointing out the potential misunderstanding. As now revised (originally in text in lines 117-119 ), “...The function $h$ represents the way in which different $p$-order ODE solvers handle the function $s$, and its specific form depends on the design of the solver. For example, in the DPM-Solver [21], an exponential integrator is used to transform $s$ into $h$ in order to eliminate linear terms. In the case of a first-order Euler-Maruyama solver [38], it serves as an identity mapping of $s$…”. We hope these modifications more accurately reflect our method and address your concerns about the potential misinterpretation.
>
> ***W3: Experiments on EDM, the SOTA diffusion model on CIFAR-10 and ImageNet 64x64.***
>
> **A**: Based on the EDM pre-trained model, we conducted experiments on CIFAR10 and ImageNet 64x64 datasets, using PFDiff with and without DDIM, as shown in the tables below:
>
> EDM, CIFAR10, FID$\downarrow$
>
> | Method\NFE  | 4         | 6         | 8        | 10       | 12       | 15       | 20       |
> | ----------- | --------- | --------- | -------- | -------- | -------- | -------- | -------- |
> | DDIM        | 73.00     | 38.36     | 24.17    | 16.55    | 12.40    | 8.89     | 6.10     |
> | DDIM+PFDiff | **58.02** | **12.60** | **4.57** | **3.15** | **2.69** | **2.39** | **2.22** |
>
> EDM, ImageNet 64x64, FID$\downarrow$
>
> | Method\NFE  | 4         | 6         | 8         | 10        | 12        | 15        | 20        |
> | ----------- | --------- | --------- | --------- | --------- | --------- | --------- | --------- |
> | DDIM        | 88.33     | 55.12     | 41.67     | 34.26     | 29.68     | 25.39     | 21.54     |
> | DDIM+PFDiff | **47.38** | **19.82** | **13.09** | **11.06** | **11.35** | **11.29** | **11.43** |
>
> As shown in the tables above, PFDiff continues to significantly improve the baseline performance of the EDM pre-trained model, further validating the effectiveness of our proposed PFDiff algorithm. We have incorporated the above experimental results into Table 4 of the revised version.
>
> ***Q1: Are there any insights why PFDiff is more effective on first-order solvers (DDIM+PFDiff even outperforms high-order solvers)?***
>
> **A**: **First order:** The effectiveness mainly comes from the efficient utilization of information and a single iteration consists of two update processes. We have conducted a thorough analysis of the algorithmic update process of PFDiff. Initially, PFDiff utilizes past gradients to replace current gradients, updating to a future state; it then calculates future gradients based on this future state; finally, it employs these future gradients to replace the current gradients, completing an iterative update cycle. In this process, when using PFDiff+DDIM (a first-order ODE solver), **the algorithm only requires one gradient computation (1 NFE) to complete two updates. This is equivalent to achieving an update process of a second-order ODE solver with 2 NFE.** Therefore, under equivalent NFE conditions, PFDiff+DDIM even surpasses high-order ODE solvers. Furthermore, we have simulated the update process of PFDiff+DDIM in Fig. 2(b), discovering significant corrections to the trajectory of the first-order ODE solver by PFDiff, which substantially increases the sampling speed of the first-order ODE solver, even exceeding that of high-order ODE solvers.
>
> However, as PFDiff introduces a small approximation bias when replacing gradients, a high-order ODE solver that calculates multiple gradients per iteration accumulates this bias. Therefore, the error accumulation when combining PFDiff with high-order ODE solvers can lead to instability under fewer NFE conditions, thereby impacting overall performance. This further illustrates that PFDiff is more suitable in conjunction with first-order ODE solvers, particularly under fewer NFE conditions.

---

> > ### Comment · Reviewer_NAJn · 2024-08-09
> >
> > Thank you for the detailed responses. My concerns are well addressed, and I keep the score of leaning towards acceptance in the current reviewing stage.

---

> > > ### Author Response · Authors · 2024-08-11
> > >
> > > We are so glad to hear that your concerns were well addressed! Thanks again for recognizing our work!

---

### Official Review · Reviewer_GukH · 2024-07-14

**Soundness:** 3
**Presentation:** 3
**Contribution:** 3
**Rating:** 6
**Confidence:** 3

**Summary:**

The paper proposes a new training-free acceleration method for the inference of diffusion probabilistic models. The key components of the presented time-skipping strategy are the use of past and future gradients to eliminate redundant neural function evaluations (NFE). The proposed method is shown effective compared to other training-free acceleration methods, leading to solid performance improvements especially for ODE solvers with less than 10 NFEs.

**Strengths:**

*    The method is training-free and can complement existing fast ODE solvers
*    The paper is well structured and puts the presented method in the proper context with respect to existing methods
*    The experimental results cover conditional and unconditional settings, showing performance improvements across the board.

**Weaknesses:**

*    The performance gap compared to training-based methods is still apparent, especially considering the latest distillation techniques resulting in one-step models.
*    The mathematical notations are a bit hard to follow up. I would advise the authors to add a schematic clarifying for a given setting of hyperparameters, which timepoints are being evaluated and which are being skipped.
*    The optimal setting of hyperparameters *k,l* is model/dataset dependent and it is not clear apriori how to set these. Therefore, this requires empirical experimentation which makes it time-consuming to get optimal performance when using the method out-of-the-box.

**Questions:**

*    How does your samples fare in terms of diversity compared to the original model with enough NFE? In other words, does the faster sampling somehow come at the cost of reduced diversity?

**Limitations:**

The authors were upfront about the limitations of their method.

---

> ### Author Rebuttal · Authors · 2024-08-06
>
> We sincerely appreciate the reviewer's recognition of our work and valuable comments. Below are our responses to all questions. We kindly hope you could consider increasing the score if you are satisfied.
>
> ***W1: The performance gap compared to training-based methods is still apparent.***
>
> **A**: Both training-based and training-free methods have their own application scenarios. While training-based acceleration algorithms can achieve one-step sampling, these methods often come with high training costs, especially when applied to large pre-trained models like Stable Diffusion. These substantial training costs of such methods significantly limit the broad applicability. In contrast, our proposed PFDiff algorithm achieves high-quality sampling under 10 NFE without any training requirement, making it a more attractive and practical solution.
>
>
>
> ***W2: The mathematical notations are a bit hard to follow up.***
>
> **A**: We deeply appreciate the valuable comments from the reviewer. To more clearly demonstrate the specific execution process of the PFDiff algorithm in a single iteration, we have added a new schematic in the appendix (see _one-page PDF attachment_, Fig. 1). In the schematic, we explain the settings of the hyperparameters and the strategy for skipping timepoints. Additionally, for some of the more complex mathematical symbols, we have provided further explanations, for example:
>
> * We have clarified the notation $x_{t_{i+1}} = \phi (Q, x_{t_{i}}, t_{i}, t_{i+1})$: This represents the update process from $t_{i}$ to $t_{i+1}$ for the current state $x_{t_{i}}$ using the ODE solver $\phi$, and leveraging the gradients $Q$ stored in buffer.
>
> * Regarding $Q \xleftarrow{\text{buffer}}  \left (  \left \\{ \epsilon\_\theta(x \_{\hat{t}\ _{n}},\hat{t}\ _{n}) \right \\}\ _{n=0}^{p-1}, t\_{i+1}, t\_{i+2}  \right )$: This denotes the process of storing gradients calculated by a $p$-order ODE solver between the intervals $ t\_{i+1}$ and $ t\_{i+2}$ into the buffer as $Q$. The set of $p$ gradients, $\left \\{ \epsilon\_\theta(x \_{\hat{t} \_{n}},\hat{t} \_{n}) \right \\} \_{n=0}^{p-1}$, encompasses values calculated between the time points $\hat{t}\_{0} = t\_{i+1}$ and $\hat{t}\_{p} = t\_{i+2}$. Specifically, for a first-order ODE solver, this process simplifies to storing the gradient at the time point $t\_{i+1}$, $\epsilon\_\theta(x \_{t \_{i+1}},t\ _{i+1}) $, into the buffer as $Q$.
>
> We hope these modifications will help the reviewer and readers better understand our method.
>
>
>
> ***W3: Regarding the issue of hyperparameter $k$, $l$ settings.***
>
> **A**: As now clarified in *common response, Q1*, our experimental results have shown some exciting outcomes; searching for $k$ and $l$ is not very time-consuming. Moreover, even without conducting a search and simply fixing $k=1$ and $l=1$, PFDiff still significantly enhances the baseline's sampling performance.
>
>
>
> ***Q1: Does the faster sampling somehow come at the cost of reduced diversity?***
>
> **A**: **Faster sampling does not reduce diversity.** The FID is a comprehensive indicator of the diversity and quality of the algorithm, while Recall is a better measure of diversity [1]. We have supplemented our experiments on the CIFAR10 dataset with the Recall metric to analyze the impact of our method on diversity. As shown in the table below:
>
> CIFAR10, Recall$\uparrow$ and FID$\downarrow$
>
> |             | Recall |  |  | FID   |  |   |
> | ----------- | ------ | ------ | ------ | ----- | ---- | ---- |
> | Method\NFE  | 10     | 20     | 1000   | 10    | 20   | 1000 |
> | DDIM        | 50.87  | 56.06  | 58.56  | 13.66 | 7.04 | 3.87 |
> | DDIM+PFDiff | **59.85**  | **59.84**  | \      | **4.57**  | **3.68** | \    |
>
> As can be seen from the table above, compared to the original model with sufficient NFE, the diversity of PFDiff does not decrease, such as 59.85 (PFDiff) Recall$\uparrow$ with 10 NFE vs. 58.56 (DDIM) Recall with 1000 NFE. This demonstrates that the faster sampling of PFDiff does not come at the expense of reduced diversity.
>
> [1] Kynkäänniemi T., et al. Improved precision and recall metric for assessing generative models, NeurIPS 2019.

---

> > ### Comment · Reviewer_GukH · 2024-08-12
> >
> > Thanks. The authors have addressed my questions, and I'm happy to retain my positive score.

---

> > > ### Author Response · Authors · 2024-08-12
> > >
> > > We are glad to know that your questions have been addressed! We greatly appreciate your valuable suggestions, which help to improve our paper. Thanks again for your recognition and for maintaining a positive score!

---

### Official Review · Reviewer_7w4h · 2024-07-16

**Soundness:** 2
**Presentation:** 2
**Contribution:** 1
**Rating:** 4
**Confidence:** 4

**Summary:**

This work proposes a training-free time step-skipping method that can be used with existing ODE solvers for reduced NFE. The method was motivated by two observations: 1) a significant similarity in the model's outputs at time step size during the denoising process and 2) a high resemblance between the denoising process and SGD. The proposed method employed gradient replacement from past time steps and rapidly updated intermediate states inspired by Nesterov momentum. The proposed method yielded promising results.

**Strengths:**

- Experimental results look promising with multiple diffusion models on diverse datasets.
- Accelerating diffusion models for sampling is an important issue and this work tried to address it.

**Weaknesses:**

- There have been a lot of prior works on accelerating diffusion models for sampling. While this manuscript cited many, it still missed important prior works - some of them look quite similar to the proposed method. Thus, the novelty of the proposed method is unclear in the current form of this manuscript. For example, using Nesterov acceleration for fast diffusion models is not really new (e.g., R Li et al., Hessian-Free High-Resolution Nesterov Acceleration For Sampling, ICML 2022). Eq (15) of this work can be seen as a special case of the following prior works such as [R1], DeepCache [28] (using past), [R3] (using three moments or future) or [R2] (using all). Some recent work like [R4] even used partial caching instead of using the whole results. A more theoretically grounded work on using Nesterov momentum for sampling can be found in [R5].
[R1] M Xia et al., Towards More Accurate Diffusion Model Acceleration with A Timestep Tuner, CVPR 2024.
[R2] A Pokle et al., Deep Equilibrium Approaches to Diffusion Models, NeurIPS 2022.
[R3] H Guo et al., Gaussian Mixture Solvers for Diffusion Models, NeurIPS 2023.
[R4] F Wimbauer et al., Cache Me if You Can: Accelerating Diffusion Models through Block Caching, CVPR 2023.
[R5] R Li et al., Hessian-Free High-Resolution Nesterov Acceleration For Sampling, ICML 2022.
- A number of acceleration works for diffusion models also investigated the feasibility of the parallel computation. Will the proposed method be parallelized for computation?
- It is unclear if the proposed method was compared with other methods in terms of computation. Will 1 NFE of the proposed method take the same computation time as 1 NFE of other methods since the proposed method contains multiple evaluations of the neural network as in Eq. (15).
- The notation and explanation are quite confusing, so it is not easy to understand the whole idea as well as the algorithm itself.

**Questions:**

See the weaknesses.

---

> ### Author Rebuttal · Authors · 2024-08-06
>
> We appreciate the reviewer's efforts and insightful comments on our work.
>
> ***W1: The novelty question of PFDiff, and its distinctions from some prior important works, such as [R1] to [R5].***
>
> **A**: We have carefully checked all the five prior works mentioned by the reviewer, and we found they are **significantly different** from our work, which further proves the novelty of our PFDiff. Following we first highlight the uniqueness and efficiency of PFDiff, and then give a detailed analysis about the difference.
>
> 1. **On the uniqueness and efficiency of PFDiff:**
>
>      + First, our novel future-gradient method is based on mean-value theorem, which is different from the Nesterov Acceleration [R5].
>      + Second, NONE of the prior works explore and evaluate the importance of the combination of both past and future gradients for the sampling acceleration. By involving both past and future gradients, **PFDiff completes two updates with just one gradient computation (1 NFE), which is equivalent to achieving an update process of a second-order ODE solver with 2 NFE.** With our tailored update process that smoothly involves the current state, as well as past and future gradients, we achieve the new state-of-the-art performance for training-free acceleration. Ablation studies (in *common response, Q3,* Table C), also now added and demonstrated the effectiveness of each component.
>
> 2. **Differences with [R1] to [R5]:**
>
>    + [R1] is a training-based accelerated sampling algorithm, which is fundamentally different from our TRAINING-FREE PFDiff. Additionally, [R1] constructs a new time-step sequence through training, which might implicitly utilize past information, but its motivation and implementation approach are significantly different from PFDiff.
>
>    + [R2] does not "**use all**" (as in [R2] Section 3.1); its update process only depends on moments BEFORE the current state (no future gradients). Moreover, the specific update process in [R2] focuses on parallel sampling while ours focuses on reducing discretization error, which is also distinctly different.
>
>    + [R3] is based on an SDE solver while our PFDiff is an ODE-based method. More importantly, [R3] optimizes sampling by estimating “first third-order” moments (training-based) of the reverse process while the PFDiff utilizes a totally different strategy as the acceleration is via the past and future gradients (TRAINING-FREE).
>
>    + [R4] and [28] partially employ cached past gradients that are different from our past gradients, which will bring more time costs compared to direct replacement.  Besides, they did not leverage future information. Our comparative experiments, detailed in the **common response, Q3, Table C**, show that incorporating future gradients with our PFDiff significantly improves sample quality and inference time compared to these methods.
>
>    + [R5] theoretically demonstrated that Nesterov can speed up the sampling of Langevin dynamics. Though our future gradients share some similarities with Nesterov’s “foresight” update, our method has four significant differences:  **First**.  [R5] does not explicitly mention that their motivation comes from “future gradients”; it only implicitly includes them in their update process.  **Second**, PFDiff's update procedure significantly differs from Nesterov’s approach as we employ a gradient replacement strategy instead of using momentum. **Third**, Based on the mean value theorem, we analyze that future gradients are more suitable for guiding the sampling of the current state, which is unrelated to the perspective of momentum in [R5]. **Last**, our experimental results (please see the *common response, Q3,* Table C) show that sampling guided solely by future gradients significantly differs in quality from PFDiff’s approach, which uses both past and future gradients for guidance.
>
> Based on the above analysis, we cited all the papers and added comparisons with [R4] and [R5]. We can conclude that PFDiff significantly differs in methodology from [R1] to [R5], and the results further demonstrate the superiority of our method.
>
>
>
> ***W2: Will PFDiff be parallelized for computation?***
>
> **A**: Potentially yes, future gradient makes this harder for PFDiff than other acceleration methods. But theoretically by using techniques like the Picard Iteration [R6], we can break the serial dependency of sampling and achieve parallelization; We will evaluate this in future work.
>
> [R6] Shih A et al., Parallel sampling of diffusion models, NeurIPS 2023.
>
> ***W3: Will 1 NFE of PFDiff take the same computation time as 1 NFE of other methods?***
>
> **A**: **Yes**, 1 NFE of PFDiff is precisely equivalent to 1 NFE of the other methods as the value of $\epsilon \_{\theta} (x\_{t_{i-(k-l+1)}}, t_{i-(k-l+1)})$ in Eq. (15) is directly retrieved from the buffer and does not require computation. Therefore, only 1 NFE computation is actually performed in Eq. (15). We added results regarding inference time, such as 1k samples, 15.79s (PFDiff) vs. 15.90s (DDIM) with 20 NFE, ( in *common response, Q3,* Table C) from which we can see that PFDiff does not introduce extra inference time.
>
> ***W4: Regarding the question of confusion in notation and explanation.***
>
> **A**: Thanks for pointing out this. As now clarified in the **common response, Q2**,  we have added more explanations of the notation and included a flowchart diagram of the PFDiff algorithm to demonstrate the update of one iteration in the **PDF attachment**.
>
> [R1] M Xia et al., Towards More Accurate Diffusion Model Acceleration with A Timestep Tuner, CVPR 2024.
>
> [R2] A Pokle et al., Deep Equilibrium Approaches to Diffusion Models, NeurIPS 2022.
>
> [R3] H Guo et al., Gaussian Mixture Solvers for Diffusion Models, NeurIPS 2023.
>
> [R4] F Wimbauer et al., Cache Me if You Can: Accelerating Diffusion Models through Block Caching, CVPR 2023.
>
> [R5] R Li et al., Hessian-Free High-Resolution Nesterov Acceleration For Sampling, ICML 2022.

---

> > ### Comment · Reviewer_7w4h · 2024-08-13
> >
> > I acknowledge that I have read the rebuttal. The rebuttal addressed almost all of my concerns. I will finalize my score after the discussion with other reviewers.

---

> > > ### Author Response · Authors · 2024-08-13
> > >
> > > Thank you very much for your response! We are glad that the rebuttal has addressed almost all of your concerns. Let us know if further discussion is needed.

---

### Author Rebuttal · Authors · 2024-08-06

Thank you to all reviewers for their efforts and valuable comments on this paper. Here we address common concerns raised by the reviewers.

## Response to common questions

***Q1: The issue regarding the hyperparameters $k$ and $l$ needs to be adjusted for different datasets/models and the NFE.***

**A**: **First**, even by directly setting $k=1$ and $l=1$ without searching, PFDiff can significantly enhance the sampling quality of the baseline across various datasets/models with 8~ 20 NFE, as shown in Table 7.  **Second**, compared to training-based acceleration algorithms, the cost of searching for $k$ and $l$ is negligible. This is because
 + Our search is training-free and can be achieved simply by image quality evaluation.
 + The evaluation is further optimized based on our **exciting discovery**: Searching using 1/10 of the data could provide a consistent searching result with the searching using the whole dataset, which largely reduces the computational cost.  For example, performance evaluation on the CIFAR10 dataset usually requires 50k samples while searching for $k$ and $l$ only need 5k samples, which get a consistent searching score with that from 50k samples.

As evidence of this observation,  we added experiments on the CIFAR10 dataset. Maintaining identical experimental settings, we search for different $k$ and $l$ combinations using 5k (randomly sampled, multiple runs) and 50k samples, respectively. As shown in Tables A and B (in the Attachment), for the same NFE, the optimal combinations of $k$ and $l$ based on the FID scores are consistent for both 5k and 50k samples. E.g., when NFE=4, both of the best FID values for both 5k and 50k samples are achieved when setting $k=3$ and $l=1$. For the six combinations used in the paper with $k \leq 3$ ($l \leq k$), only a total of 30k samples are required to search the optimal $k$ and $l$ combination for each NFE. This is even less than the cost of EVALUATION (normally using 50k samples). This observation is also observed across other datasets/models, and we have added additional experiments with reduced sampling sizes for more datasets/models after Appendix Table 7.


***Q2: The issue of unclear explanations regarding the algorithmic process and the symbols used.***

**A**: Following the reviewers' valuable suggestions, we further clarified the PFDiff method by introducing a flowchart as an illustration of the algorithm (see Figure 1 in the Attachment).
In addition to the schematic, we have provided further explanations for the symbols in the revised version of the paper. For instance:

* We have clarified the notation $x_{t_{i+1}} = \phi (Q, x_{t_{i}}, t_{i}, t_{i+1})$: This represents the update process from $t_{i}$ to $t_{i+1}$ for the current state $x_{t_{i}}$ using the ODE solver $\phi$, and leveraging the gradients $Q$ stored in buffer.

* Regarding $Q \xleftarrow{\text{buffer}}  \left (  \left \\{ \epsilon\_\theta(x \_{\hat{t}\ _{n}},\hat{t}\ _{n}) \right \\}\ _{n=0}^{p-1}, t\_{i+1}, t\_{i+2}  \right )$: This denotes the process of storing gradients calculated by a $p$-order ODE solver between the intervals $ t\_{i+1}$ and $ t\_{i+2}$ into the buffer as $Q$. The set of $p$ gradients, $\left \\{ \epsilon\_\theta(x \_{\hat{t} \_{n}},\hat{t} \_{n}) \right \\} \_{n=0}^{p-1}$, encompasses values calculated between the time points $\hat{t}\_{0} = t\_{i+1}$ and $\hat{t}\_{p} = t\_{i+2}$. Specifically, for a first-order ODE solver, this process simplifies to storing the gradient at the time point $t\_{i+1}$, $\epsilon\_\theta(x \_{t \_{i+1}},t\ _{i+1}) $, into the buffer as $Q$.

***Q3: The issue concerning the effectiveness and inference time of PFDiff.***

**A**: The efficiency of the PFDiff algorithm is attributed to its **information-efficient** update process that utilizes the current intermediate state along with past and future gradients. Omitting either past or future gradients when updating the current intermediate state will significantly limit the effectiveness of PFDiff. Based on the experimental setup in Table 8, we added more ablation studies of PFDiff to evaluate the effectiveness of past or future gradients and also introduced an additional comparison method [1] as it utilizes a portion of the past gradient cache to accelerate sampling. We also reported the inference time at 10 and 20 NFE (on an NVIDIA RTX 3090 GPU) to demonstrate the efficiency of different methods. The results are shown in the table below (also in the revised manuscript, Table 8):

**Table C**. CIFAR10

|                | FID↓ |       |        |        | Time per 1k samples (s)↓ |        |
| -------------- | --------------- | ----- | ------ | ------ | ----------------------------------- | ------ |
| **Method\NFE** | **4**           | **8** | **10** | **20** | **10**                              | **20** |
| DDIM           | 65.70           | 18.45 | 13.66  | 7.04   | 9.81                                | 15.90  |
| +Cache [1]     | 49.02           | 15.23 | 11.31  | 6.25   | 13.55                               | 24.07  |
| +Past          | 52.81           | 17.87 | 13.64  | 7.02   | 9.88                                | 15.81  |
| +Future        | 66.06           | 11.93 | 8.06   | 4.07   | 9.77                                | 15.67  |
| +PFDiff        | 22.38           | 5.64  | 4.57   | 3.68   | 9.74                                | 15.79  |

As can be seen from the table, using only past gradients (including cache) or only future gradients does not effectively accelerate sampling. Therefore, the tailored update of the PFDiff that combines both the past and future gradients works as the key factor in its effectiveness. Additionally, we compared DDIM and DDIM+PFDiff in terms of inference time per 1k samples and found that both have consistent inference times at the same NFE, demonstrating that PFDiff does not increase inference time.

[1] Xinyin Ma et al., Deepcache: Accelerating diffusion models for free, CVPR 2024.

---

### Decision · Program_Chairs · 2024-09-25

**Decision:**

Reject

**Comment:**

This work addressed the common discretization errors of fast ODE solvers in accelerating the sampling of Diffusion Probabilistic Models. These errors typically  grow as the number of function evaluations decreases, and the  proposed method, namely PFDiff, is presented as a training-free with an  orthogonal timestep-skipping strategy, which effectively reduces the number of function evaluations relative to existing fast ODE solvers. The strategy is specifically rooted in exploiting two key observations: a strong closeness of  the model's outputs at relatively close time step size during the denoising process of existing ODE solvers, and a resemblance between the denoising process and SGD. The proposed PFDiff uses a  gradient replacement from past time steps and foresight updates to  rapidly update intermediate states, and  reduce unnecessary NFE which affect the discretization errors in first order ODEs. A solid experimental evaluation validates the claimed speed improvement , demonstrating applicability across a number pre-trained Diffusion Probabilistic Models, to surpass methods for   conditional DPMs and  previous state-of-the-art training-free methods.
Five reviews were  submitted with 2 W. Accpt, 1 BL Accpt. and 2 BL. Rejects. The authors  have been very meticulous in their rebuttal with additional experiments and comparisons, and point to point rebuttal: In particular, Reviewer 7w4h claimed no novelty and provided  references, the authors have addressed each and every reference showing that the reviewer was either confused or not totally familiar with the references provided. They have also provided answers to all the questions to the Reviwer's satisfaction  and who had promised to revisit the grade. Well he did not.  The other BL Reject by Reviwer FcEN has raised a number of points most of which turned out to be the result of his/her  confusion. Specifically, FcEN asking for further explanations why using past gradients was helpful in guiding the new faster sampling approach, which of course was not the case and the authors politely pointed that out. Having requested an evaluation and comparison based on ImageNet 256, with an openness to review his grading. All that was provided by the authors with no follow up with several reminders sent by the AC.
The AC is reluctantly ranking this paper as reject in light of the two "weak" BRs, The AC quickly read the paper and read the additional edits proposed by the authors, and estimates that the authors did their part, and while the paper is perhaps not award winning, it is acceptable as a Poster.